# ⚀⚄ *Roll the dice & look before you leap*:
# Going beyond the creative limits of next-token prediction

Vaishnavh Nagarajan [* 1]   Chen Henry Wu [* 2]   Charles Ding [2]   Aditi Raghunathan [2]

## Abstract

We design a suite of minimal algorithmic tasks that are a loose abstraction of *open-ended* real-world tasks. This allows us to cleanly and controllably quantify the creative limits of the present-day language model. Much like real-world tasks that require a creative, far-sighted leap of thought, our tasks require an implicit, open-ended *stochastic* planning step that either (a) discovers new connections in an abstract knowledge graph (like in wordplay, drawing analogies, or research) or (b) constructs new patterns (like in designing math problems or new proteins). In these tasks, we empirically and conceptually argue how next-token learning is myopic; multi-token approaches, namely teacherless training and diffusion models, comparatively excel in producing diverse and original output. Secondly, to elicit randomness without hurting coherence, we find that injecting noise at the input layer (dubbed *seed-conditioning*) works surprisingly as well as (and in some conditions, better than) temperature sampling from the output layer. Thus, our work offers a principled, minimal test-bed for analyzing open-ended creative skills, and offers new arguments for going beyond next-token learning and temperature sampling. We make part of the code available under https://github.com/chenwu98/algorithmic-creativity

## 1. Introduction

Not all forms of intelligence are solely about being correct or wrong. In *open-ended* tasks, what also matters is finding creative ways to satisfy a request, making surprising and fresh connections never seen before. For instance, consider responding to highly under-specified prompts like "`Generate a challenging high-school word problem involving the Pythagoras Theorem.`" or "`Suggest some candidate therapeutic antibodies targeting the HER2 antigen.`" or "`Provide a vivid analogy to differentiate quantum and classical mechanics.`" Creativity in such tasks requires generating responses that are not just *correct* or *coherent*, but are also *diverse* across responses and are *original* compared to the training data. These currently-sidelined desiderata will rise to prominence as we explore LLMs for open-ended scientific discovery (Gruver et al., 2023; Romera-Paredes et al., 2024; Hayes et al., 2025), for generating novel training data (Yu et al., 2024; Yang et al., 2024c; Wang et al., 2023), and as we scale up test-time compute approaches that benefit from diversity in exploration, such as best-of-N (Cobbe et al., 2021; Chow et al., 2024; Dang et al., 2025) and long chain-of-thought reasoning (OpenAI, 2024; DeepSeek-AI, 2025; Snell et al., 2024; Wu et al., 2024).

Unlike simple open-ended tasks like generating names and basic sentences (Zhang et al., 2024b; Hopkins et al., 2023; Bigelow et al., 2024), many creative tasks (like designing a clever Olympiad problem) are said to involve a random flash of creative insight termed variously as a *leap of thought* (Wang et al., 2024a; Talmor et al., 2020; Zhong et al., 2024), a "eureka" moment (Bubeck et al., 2023), a mental leap (Holyoak & Thagard, 1995; Callaway, 2013; Hofstadter, 1995) or an incubation step (Varshney et al., 2019). The thesis of this paper is that learning to solve such creative leap-of-thought tasks (defined shortly) is misaligned with the current language modeling paradigm (a) in terms of next-token learning, and (b) in how randomness is elicited. We articulate these two concerns by designing a suite of algorithmic tasks inspired by such creative tasks. We then demonstrate how the creativity of language models suffers in these tasks, and how this can be alleviated (to an extent, within our tasks).

Concretely, for the scope of this paper, a creative leap-of-thought task refers to tasks that involve a search-and-plan process; crucially, this process orchestrates multiple *random*

---

*Equal contribution [1]Google Research, US [2]Carnegie Mellon University, Pittsburgh, US. Correspondence to: Vaishnavh Nagarajan <vaishnavh@google.com>, Chen Henry Wu <chenwu2@cs.cmu.edu>.

*Proceedings of the 42nd International Conference on Machine Learning*, Vancouver, Canada. PMLR 267, 2025.

decisions *in advance* before generating the output. Typically, such a leap of thought is *highly implicit* in the text — to infer it, one has to deeply engage with the text and detect higher-order patterns. We can think of tasks like designing satisfying math problems, generating worthwhile research ideas, or drawing surprising analogies as examples of such tasks.

Ideally, one would directly study these real-world tasks to quantify the limits of language models. Indeed, a flurry of recent works report that LLM-generated research ideas tend to be rephrased from existing ideas (Gupta & Pruthi, 2025; Beel et al., 2025) and that LLM outputs tend to be less creative than humans e.g., Chakrabarty et al. (2024); Lu et al. (2024b) (See §K). While assessing real-world tasks is a lofty goal, the evaluations are subjective (Wang et al., 2024b; Runco & Jaeger, 2012), and when the model has been exposed to all of the internet, originality is hard to ascertain. Thus, the conclusions will inevitably invite debate (such as Si et al. (2024) vs. Gupta & Pruthi (2025) or Lu et al. (2024a) vs. Beel et al. (2025)).

In search of more definitive conclusions, we approach from a different angle: we study controllable tasks that are loose abstractions of real-world tasks and yet allow one to objectively quantify originality and diversity. This follows recent works that have studied diversity of models in path-finding (Khona et al., 2024) and challenging context-free grammars (CFGs) (Allen-Zhu & Li, 2023b). We broadly term these as open-ended algorithmic tasks. Our aim then is design minimal instances of these tasks, akin to basic arithmetic for reasoning, so as to tease apart the bare minimum computational skills required for creativity. Then, we can examine how the current modeling paradigm may be limited even with this basic skills, and what alternative paradigms may be desired.

As our first main contribution, we draw inspiration from cognitive science literature (Boden, 2003) (see also Franceschelli & Musolesi (2023)) to design algorithmic tasks isolating two distinct types of creative leaps of thought. The first class of tasks involves *combinational creativity*: drawing novel connections in knowledge, like in research, wordplay or drawing analogies (see Fig 1 for task description). The second class of tasks involves *exploratory creativity*: constructing fresh patterns subject to certain rules, like in designing problems and suspense (see Fig 2). In these tasks, we can precisely evaluate models for the fraction of generations that are *coherent, unique and original* (not present in training set). We term this metric "algorithmic creativity" to denote that it solely evaluates the computational aspects of creativity.

Within this framework, we articulate two creative limits of the current language modeling paradigm. First, we empirically find that next-token learning achieves lower algo-

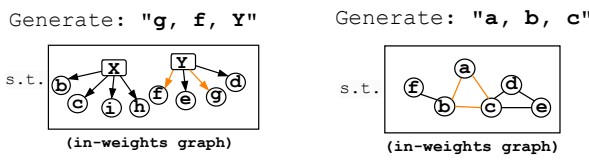

(a) `Sibling Discovery`    (b) `Triangle Discovery`

*Figure 1.* **Minimal tasks inspired by combinational creativity:** Skills like research, humor and analogies often require identifying novel multi-hop connections from *known* pair-wise relationships in a knowledge graph. For instance, creating the wordplay "`What kind of shoes do spies wear? Sneakers.`" requires searching over a semantic graph, and carefully planning a pair of words (`shoes`, `spies`) that lead to a mutual neighbor (`sneakers`). Inspired by this, we define tasks where a symbolic graph is stored in the model weights; the model is exposed to example node sequences that form a specific multi-hop structure (like a sibling or a triangle) during training. The model must infer this structure from training; during inference, the model must implicitly recall-search-and-plan to generate novel and diverse node sequences obeying the same structure in the in-weights graph. Pictured are two example tasks with a symbolic graph each, and a corresponding example sequence obeying a sibling (`g`, `f`, `Y`) or a triangle structure (`a`, `b`, `c`). More details in §2.3 and Fig 9.

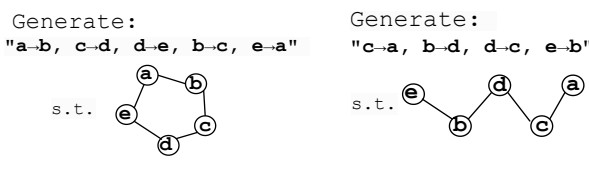

(a) `Circle Construction`    (b) `Line Construction`

*Figure 2.* **Minimal tasks inspired by exploratory creativity:** Skills like designing problem sets, novel proteins and plots require devising patterns that can be *resolved* in novel ways through some general rules. Inspired by this, we design a task where during training, we expose the model to "adjacency lists" that implicitly resolve into a specific structure (a circle or a line graph) under some permutation. The model must infer this higher-order structure; during inference, the model must generate adjacency lists resolving to the same structure, but under novel and diverse permutations. Pictured are example sequences and the corresponding implicit structure they would resolve to. See §2.4 and Fig10.

rithmic creativity compared to some multi-token approach, namely, either teacherless training (Bachmann & Nagarajan, 2024; Monea et al., 2023; Tschannen et al., 2023) or diffusion (Hoogeboom et al., 2021; Austin et al., 2021; Lou et al., 2023) (see Fig 3 and Fig 4). Our argument is that in all our tasks, inferring the latent leap of thought requires observing global higher-order patterns rather than local next-token patterns in the sequence.

Next, we turn to how we elicit randomness from a Transformer. While the de facto approach is to elicit randomness

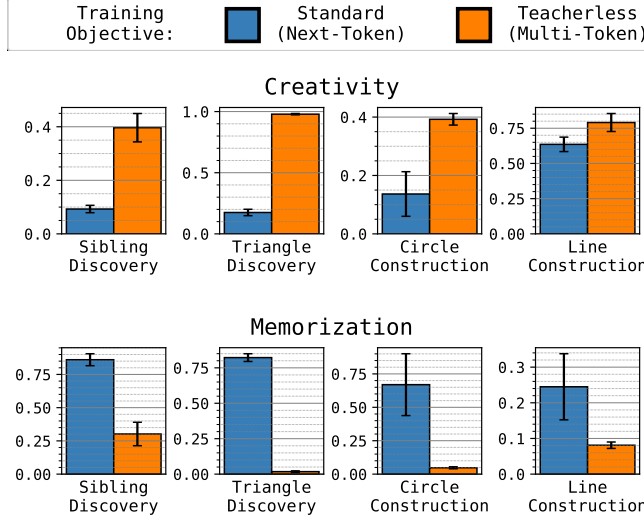

*Figure 3.* **Multi-token teacherless finetuning improves algorithmic creativity (top; Eq 1) and reduces memorization (bottom; fraction of generations seen during training) on our four open-ended algorithmic tasks for a `Gemma v1 (2B)` model**.

at the output — temperature sampling — we contrast this with injecting randomness into the model. Concretely, we study *seed-conditioning*, where we train and test with a random prefix string per example. Surprisingly, we find that seed-conditioning induces non-trivial algorithmic creativity even with deterministic decoding; it is in fact, comparable to temperature sampling (and in some cases, better). We intuit that maximizing diversity at the output-level results in cognitive overload: it requires simultaneously processing many leaps of thoughts to compute a marginalized token distribution. Input-level noise injection could sidestep this by articulating one leap of thought per seed. We also view this exploration as a controlled study solidifying prior indications that prompt variations induce diversity (Li et al., 2023; Lau et al., 2024; Naik et al., 2024; Li et al., 2022).

Overall, we hope to advance the field in two directions. First, we provide a new angle to advocate for multi-token approaches, orthogonal to the "path-star" example in Bachmann & Nagarajan (2024) (or B&N'24 in short). Whereas, the path-star example portrays a gap in correctness of reasoning, ours shows a gap in diversity of open-ended thinking. We note though that B&N'24 is an impossibility result where next-token learning breaks down spectacularly (unless there is exponential data or compute), while ours is a data-inefficiency result (where next-token learning occurs but is mediocre). Next, the gap we show appears even in 2-token-lookahead tasks as against the many-token-lookahead path-star task. Third, and most conceptually important is the fact that, while the path-star task is amenable to next-token prediction upon reversing the tokens, we identify tasks where no re-ordering is friendly towards next-token

prediction — *the optimal thing to do is to globally learn higher-order patterns implicit in the whole future sequence.* This presents a challenge to recent proposals that aim to fix next-token prediction via permutations (Pannatier et al., 2024; Thankaraj et al., 2025) or partial lookaheads (Bavarian et al., 2022; Fried et al., 2022; Kitouni et al., 2024; Nolte et al., 2024).

As a second direction of progress, we hope our work provides a foundation to think about open-ended tasks which are extremely hard to quantify in the wild. This may spur more algorithmic explorations on improving diversity (such as our approach of seed-conditioning) and on curbing verbatim memorization in language models.

**Our contributions:**

1. We create minimal, controlled and easy-to-quantify open-ended algorithmic tasks. These tasks isolate, and loosely capture two fundamental modes of creativity.

2. We find that multi-token prediction through one of teacherless training or diffusion, results in significantly increased algorithmic creativity in our tasks compared to next-token prediction.

3. Our argument provides new support for multi-token prediction, going beyond B&N'24. We show a gap in creativity in an open-ended task (rather than correctness in a deterministic one), in much simpler 2-token-lookahead tasks, and in tasks where no token permutation is friendly to next-token-learning.

4. We find that input-randomization via seed-conditioning achieves algorithmic creativity comparable with conventional output-randomization via temperature sampling.

## 2. Open-ended algorithmic tasks & two types of creativity

We are interested in designing simple algorithmic tasks that are loosely inspired by endeavors such as generating scientific ideas, wordplay, narration, or problem-set design, where one needs to generate strings that are both "interesting" and never seen before. In all these tasks, before generating the output, one requires *a (creative) leap of thought*, a process that (a) is implicit i.e., is not spelled out in token space (or is even inherently hard to spell out), (b) involves discrete *random* choices (c) and together, those choices must be *coherent* in that they are carefully planned to satisfy various non-trivial, discrete constraints. These constraints fundamentally define the task and make it interesting e.g., a word problem should be solvable by arithmetic rules, or a pun must deliver a surprising punchline. The goal in such open-ended tasks is not just coherence though, but also diversity and novelty — generations must be as varied as possible and must not be regurgitated training data.

Before we design tasks that require the aforementioned leap of thought, we first clarify what tasks do not require it.

**Open-ended tasks that do *not* require a leap of thought.** One simple open-ended task that may come to mind is generating uniformly-random known entities, like celebrity names (Zhang et al., 2024b). However, there is no opportunity to create a novel string here. A more interesting example may be generating grammatically coherent probabilistic CFG strings following a `subject verb object` format e.g., `the cat chased a rat` (Hopkins et al., 2023). While novel strings become possible now, no sophisticated leaps of thought are involved; each token can be generated on the fly, satisfying a local next-token constraint to be coherent. In light of this, we can rephrase our goal as designing open-ended, creative tasks where coherence requires satisfying more interesting, "global" constraints. To build this systematically, we draw inspiration from literature in cognitive science (Boden, 2003). Boden (2003) argues that fundamentally, there are three forms of creativity in that order: *combinational, exploratory* and *transformative*. We elaborate on the first two (the last, we do not look at).

### 2.1. The fundamental types of creativity (Boden, 2003)

**Combinational creativity.** Consider rudimentary word-play of the form "What musical genre do balloons enjoy? Pop music." or "What kind of shoes do spies wear? Sneakers." There is a global structure here: two unrelated entities (`genre` & `balloons`) are related eventually through a punchline (`pop`); the punchline itself is a mutual neighbor on a semantic graph. More broadly, Boden (2003) argues that many tasks, like the above, involve "making unfamiliar combinations of familiar ideas" or the "unexpected juxtaposition of [known] ideas". Other tasks include drawing analogies, or finding connections between disparate ideas in science. All these tasks involve a leap of thought that in effect searches and plans over a space of known facts and combines them.

**Exploratory creativity.** Consider the act of developing a mystery or designing logical puzzles. These endeavors are not as knowledge-heavy. What they crucially require is constructing fresh patterns that satisfy some highly non-trivial global constraint e.g., being resolvable as per some rules (e.g., logic). Such endeavors fall into a second class of *exploratory* creativity in Boden (2003). This includes much grander forms of exploration e.g., exploring various forms of outputs within a stylistic constraint, or exploring various corollaries within a theoretical paradigm in physics or chemistry. The leap of thought here requires searching over all possible sequences, constrained by a set of rules.

In the upcoming sections, we will capture some core computations of rudimentary instances within the two creative skills above. By no means does this minimal algorithmic

setup intend to capture the human values or the subjective aspects of these endeavors (Schmidhuber, 2009); nor do they capture the rich array of creative acts that Boden (2003) discusses within these categories. (See limitations in §A).

### 2.2. The basic setting and notations

In all our tasks, we assume the standard generative model setting: the model must learn an underlying distribution $\mathcal{D}$ through a training set $S$ of $m$ independent samples $s_i \sim \mathcal{D}$. The distribution is over a space $\mathbb{V}^L$ of $L$-length strings composed of tokens from the vocabulary $\mathbb{V}$. The tasks are open-ended in that there is no one correct answer at test-time. The goal *is* to produce a random string from $\mathcal{D}$, much like responding to the query `Design a high-school word problem`. We hope that this setup loosely resembles pretraining for language or in applications like drug discovery or protein (Meier et al., 2021; Madani et al., 2023; Watson et al., 2023) and genome modeling (Rives et al., 2021; Nguyen et al., 2024), where the model is trained on a set of varied example sequences, and is later used to produce novel outputs during inference.

**Coherence:** Each task is defined by a boolean coherence function $\mathtt{coh} : \mathbb{V}^L \mapsto \{\mathtt{true}, \mathtt{false}\}$ which is true only on the support i.e., $\mathtt{supp}(\mathcal{D}) = \{s \in \mathbb{V}^L | \mathtt{coh}(s)\}$. The exact form of $\mathtt{coh}$ will be defined in each algorithmic task but broadly, we are interested in scenarios where determining coherence requires a global understanding of the whole string. This is inspired by the fact that a wordplay must have a preplanned punchline connecting what comes before, or a word problem must be solvable. We can think of $\mathcal{D}$ to be a simple uniform distribution over all coherent strings.

**Algorithmic creativity:** Upon witnessing a finite set of examples, the model must learn to generate only strings that are (a) coherent, (b) original (not memorized) and (c) diverse (covers the whole support). An exact quantification of this is computationally expensive in our tasks. Instead, we approximate it by sampling a set $T$ of many independent generations from the model and computing the fraction of $T$ that is original, coherent and unique. Let the boolean $\mathtt{mem}_S(s)$ denote whether an example $s$ is from the training set $S$ and let the integer function $\mathtt{uniq}(X)$ denote the number of unique examples in a set $X$. (The exact definitions of these quantities vary by tasks, as we will see). Then, we define our (empirical) algorithmic creativity metric:

$$\hat{\mathtt{cr}}_N(T) = \frac{\mathtt{uniq}(\{s \in T | \neg \mathtt{mem}_S(s) \wedge \mathtt{coh}(s)\})}{|T|}. \quad (1)$$

Admittedly, we are looking at a simple form of novelty that is in-distribution. This notion, while simple, finds its relevance in applications like modeling proteins and genomes. We will also see that this notion is non-trivial enough to demarcate the limits of language models.

## 2.3. Tasks inspired by combinational creativity

Combinational creativity requires a recall-and-search through entities from memory, constrained to relate with each other in an interesting way. We abstract this through tasks that discover structures from an *in-weights* graph i.e., a graph stored in the model weights, not reveal in context.

### 2.3.1. SIBLING DISCOVERY

This task involves an implicit, bipartite $\mathcal{G}$ with parent vertices $\mathcal{V} = \{A, B, C, \ldots\}$ each neighboring a set of children $\mathtt{nbr}(A) = \{a_1, a_2, \ldots, \}$, $\mathtt{nbr}(B) = \{b_1, b_2, \ldots\}$ and so on. We define $\mathtt{coh}(\boldsymbol{s})$ to be true on sibling-parent triplets of the form $\boldsymbol{s} = (\gamma, \gamma', \Gamma)$ such that $\gamma, \gamma' \in \mathtt{nbr}(\Gamma)$. We then consider a uniform distribution $\mathcal{D}$ over all coherent strings $(\gamma, \gamma', \Gamma)$ for a fixed graph $\mathcal{G}$. The model witnesses i.i.d samples from $\mathcal{D}$. During test-time, the model must maximize algorithmic creativity (Eq 1) by generating novel parent-sibling triplets based on its in-weights knowledge of $\mathcal{G}$. Note that the model is not provided the graph in-context as this would sidestep a core computational step in combinational creativity: recalling facts from a large memory (see §C.4). The hope is that the model infers and stores the pairwise adjacencies of $\mathcal{G}$ in its weights (given sufficient data). Full dataset description is in §D and Fig 9.

We view this task as an abstraction of the wordplay example. One can think of the parent $\Gamma$ as the "punchline" that delivers a connection between otherwise non-adjacent vertices, in the same way $\mathtt{sneaker}$ surprisingly connects the otherwise non-adjacent words, $\mathtt{spies}$ and $\mathtt{shoes}$.

**What is a leap of thought?** The natural order of generation is to pick the parent vertex (i.e., punchline) first, and the siblings after (conditioned on the parent). Thus, if the task demanded producing $(\Gamma, \gamma, \gamma')$, the tokens we observe betray the way they are naturally generated. However, the wordplay example presents the punchline (the parent, $\Gamma$) last while implicitly it must be planned ahead. We term this implicit (random) planning step a leap of thought. Paralleling this, we define our sibling discovery task to generate the triplets as $\boldsymbol{s} = (\gamma, \gamma', \Gamma)$, where the parent appears last. We hypothesize that this (sibling-first) construction is adversarial towards next-token learning, while a reversed (parent-first) dataset is friendlier towards next-token learning. (More on this in §2.6.)

### 2.3.2. TRIANGLE DISCOVERY

Next, we design a task that requires a more complex, higher-order planning: generating triangles from an appropriately-constructed knowledge graph $\mathcal{G} = (V, E)$ (see graph construction in §D). Thus, in this task $\mathtt{coh}((v_1, v_2, v_3)) = \mathtt{true}$ iff all three edges between $\{v_1, v_2, v_3\}$ belong in $\mathcal{G}$. Furthermore, we define $\mathtt{uniq}(\cdot)$ and $\mathtt{mem}(\cdot)$ such that vari-

ous permutations of the same triangle are counted as one (see details in §D, including the exact formatting of the string). Note that the leap of thought in this task is much harder to learn and execute as it requires co-ordinating three edges in parallel, from memory.

This type of a higher-order planning task can be thought of an abstraction of more complex wordplay (like *antanaclasis*, where a word must repeat in two different senses in a sentence, while still being coherently related to the rest of the sentence) or creating word games (like crosswords) or discovering contradictions or feedback loops in a body of knowledge, an essential research skill — see §C.5.

## 2.4. Tasks inspired by exploratory creativity

We are also interested in the less-knowledge-intensive creativity involved in tasks like designing word problems that demand novel solutions. We capture this skill minimally through tasks that construct new structures. No knowledge graph is involved in these tasks.

### 2.4.1. CIRCLE CONSTRUCTION

In this task, the model must generate adjacency lists that can be rearranged to recover circle graphs of $N$ vertices. Let the generated list be $\boldsymbol{s} = (v_{i_1}, v_{i_2}), (v_{i_3}, v_{i_4}), \ldots$. We define $\mathtt{coh}(\boldsymbol{s}) = \mathtt{true}$ iff there exists a *resolving* permutation $\pi$ such that $\pi(\boldsymbol{s}) = (v_{j_1}, v_{j_2}), (v_{j_2}, v_{j_3}), \ldots (v_{j_n}, v_{j_1})$ for distinct $j_1, j_2, \ldots j_n$. i.e., each edge leads to the next, and eventually circles back to the first vertex. We define $\mathtt{uniq}$ and $\mathtt{mem}$ such that different examples with the same resolving $\pi$ are counted as the same, even if they have differing vertices. As always, the learner is then exposed to a finite set of uniformly sampled coherent strings. Note that the latent leap of thought here requires constructing a novel permutation $\pi$ before generating the sequence.

Loosely, we can think of the resolving permutation $\pi$ as *how* a conflict in a story or a word problem or a puzzle is solved; the vertices as characters or mathematical objects; the rules of rearranging an adjacency list as rules of logic, math or story-building. The creative goal in this task is to create novel dynamics in the conflict, or equivalently, novel dynamics in how the conflict is resolved. Thus if only the entities differ, but the plot dynamics remain unaltered, we count them as duplicates. See details in §D.

### 2.4.2. LINE CONSTRUCTION

A simple variant of the above task is one where the edge list is of a line graph. The resolving permutation $\pi$ satisfies $\pi(\boldsymbol{s}) = (v_{j_1}, v_{j_2}), (v_{j_2}, v_{j_3}) \ldots, (v_{j_{n-1}}, v_{j_n})$ for distinct $j_1 \ldots j_n$. i.e., each edge leads to the next until a dead-end.

## 2.5. Leap of thought is obscured at the token-level

Rarely does human-written creative output come annotated with an articulation of the background creative process. Nor does each protein or molecule come with an enumeration of the laws that generated it — in fact, it is because those laws are not fully known do we delegate the task of inferring those to a machine. Likewise, in our setting, the training signal consists only of the observed tokens; the model must infer the underlying leap of thought.

In fact, in our last three tasks, the model must infer a deeper form of implicit structure than what exists in many typical algorithmic tasks, like addition (Lee et al., 2024), or the path-star task (B&N'24) or Sibling Discovery. Whereas in those tasks, the leap of thought is perceptible at the token-level, modulo some re-ordering, our last three tasks have no such token ordering. These task are permutation-invariant — no token is more privileged than the other, and all tokens must be "simultaneously learned" to infer the underlying process. We view this as an abstraction of tasks where the creative process is not immediate from the surface of the text. These tasks offer a test-bed even for non-next-token approaches that rely on re-permuting the tokens (Pannatier et al., 2024; Thankaraj et al., 2025) or predicting only a part of the future (Kitouni et al., 2024; Nolte et al., 2024; Bavarian et al., 2022; Fried et al., 2022).

## 2.6. How next-token learning may suffer in our tasks

Much like in sophisticated creative tasks, in our tasks, the most natural way to generate the string is by planning various random latent choices (say $z$) in advance and by producing a *plan-conditioned* distribution $p(s|z)$ over coherent strings $s$. However, next-token prediction (NTP) – or next-token learning to be precise – we argue, is myopic and may struggle to learn such a latent plan. Our argument extends that of B&N'24 to our even simpler tasks.

Consider the sibling task where we must generate sibling-parent triplets $(\gamma, \gamma', \Gamma)$. The most natural generative rule is to plan the last token (the parent) first and decide the children last. Think of this as learning a latent plan $z := \Gamma$. Then, learning the plan-conditioned generator $p(\gamma, \gamma', \Gamma|z)$ factorizes to learning the distribution of children conditioned on a parent as $p(\gamma|z := \Gamma)$ and $p(\gamma'|z := \Gamma)$ (due to conditional independence), and the trivial $p(\Gamma|z := \Gamma)$. This requires only as many parent-sibling edges as there are in the graph, i.e., $O(m \cdot n)$ many points, if there are $m$ parents, each with $n$ children. This is optimal.

Things should proceed differently with NTP. We argue that a NTP-learner would fail to learn the plan $z := \Gamma$. The key intuition is that an NTP-learner learns the parent $\Gamma$ witnessing the siblings $(\gamma, \gamma')$ as input. This is trivial to fit: the parent is simply the mutual neighbor of the two siblings revealed

in the prefix; B&N'24 term this a "Clever Hans cheat" since the model witnesses and exploits part of the ground-truth it must generate (the siblings). Such cheats, being simpler than even the true generative rule, are quickly picked up during learning due to the well-known simplicity bias of gradient-descent-trained neural networks (Shah et al., 2020). Subsequently, any gradient supervision from the parent $\Gamma$ is lost (also known as gradient starvation (Pezeshki et al., 2021)), leaving no guidance to learn the latent plan, $z := \Gamma$.

After the Clever Hans cheat is learned, we conjecture that the NTP-learner learns the second sibling not through the plan-conditioned distribution $p(\gamma'|z := \Gamma)$ but through the next-token-conditional, $p(\gamma'|\gamma)$. This is a complex distribution: learning this may require witnessing every sibling-sibling pair totalling $O(m \cdot n^2)$ many training data — larger by a factor of $n$ compared to the natural rule.

**Summary of argument.** Abstractly, in a creative leap-of-thought task, it is most efficient to learn a well-planned random latent $p(z)$ and a subsequent latent-conditioned distribution $p(s|z)$. However, NTP factorizes this into pieces of the form $p(s_i|s_{<i}, z)$. Consequently, on the later tokens, the model may be lured by Clever Hans cheats and learn uninformative latents. Conversely, the model may learn the earlier through complex rules, bereft of a latent plan. While this may not lead to complete breakdown of learning as in B&N'24 in our tasks, we hypothesize it must lead to data-hungry learning.

## 3. Training and Inference

**Transformers.** For our next-token-trained (NTP) models, we use the standard teacher-forcing objective used in supervised finetuning. Given prompt $p$ and ground truth sequence $s$, the model is trained to predict the $i$'th token $s_i$, given as input the prompt and all ground truth tokens up until that point, $(p, s_{<i})$. We write the objective more explicitly in §B Eq 2. For the multi-token Transformer models, we use teacherless training (Monea et al., 2023; Bachmann & Nagarajan, 2024; Tschannen et al., 2023), where the model is trained to predict $s_i$ simultaneously for all $i$, only given the prompt $p$ (and some dummy tokens masking the $s$ that was once given as input). Since the exact details of this is irrelevant to our discussion, we describe this in Eq 2. To train our models, we use a hybrid of this objective and the next-token objective.

**Diffusion models.** We use the score entropy discrete diffusion model (SEDD (90M), Lou et al., 2023). Loosely, we can think of the objective here as a hybrid of the teacherless objective, where the input is fully masked or corrupted (and the model must unmask or uncorrupt all of it), and various easier objectives with partial maskings or corruptions of the input. During inference, the model starts with a fully

masked or corrupted sequence of tokens, and iteratively corrects subsets of them.

**Inference.** We extract *independent* samples from the model. For Transformers, during inference, we perform standard autoregression in both the next- and multi-token trained settings. We do this either with greedy decoding or with nucleus sampling (Finlayson et al., 2024).

### 3.1. Seed-conditioning

While temperature sampling is the standard way to elicit diversity from a Transformer, we speculate that this can lead to cognitive overload: the model must process multiple thoughts to compute a diverse softmax distribution. Alternatively, we propose conditioning on a random seed as input to the model, hoping that a thought can be fleshed out with singular focus. Thus, during training, each point is associated with an arbitrary (meaningless) prefix (e.g., of random characters) called a seed (note that there is no prefix in our tasks besides the seed). During test-time, novel seeds are used to extract test data. We can also view seed-conditioning more neutrally as a controlled way to simulate variations in prompt-wordings for a fixed task (e.g., `design a word problem` vs `construct a problem`). We elaborate on these intuitions in §C.1.

## 4. Experimental results

**Key details.** Part of our experiments are performed for a `Gemma v1 (2B)` pre-trained model (Gemma Team et al., 2024), averaged over 4 runs. For diffusion, we use a 90M (non-embedding) parameters Score Entropy Discrete Diffusion model (`SEDD (90M)`; Lou et al., 2023). For a fair comparison against NTP, we use a 86M (non-embedding) parameters `GPT-2 (86M)` model (Radford et al., 2019). In all our experiments, we finetune the models until it is clear that algorithmic creativity (Eq. 1) has saturated. All values are reported from this checkpoint. Finally, since our best Transformer results were under seed-conditioning (for both next- and mult-token training), our main results are reported under that training setting; we provide various ablations without that as well. Please see §E for more experimental details, and §D for precise dataset details (e.g., how the graph is constructed, how sequences are formatted etc.,).

### 4.1. Observations

**Multi-token prediction improves algorithmic creativity significantly.** In all our datasets, the algorithmic creativity of the `Gemma v1 (2B)` model increases significantly under multi-token prediction ( Fig 3), with nearly a 5x factor for the discovery datasets. Note that for this, we have selected the learning rate favorable towards next-token prediction; tuning for multi-token yields further gains (Fig 15). For

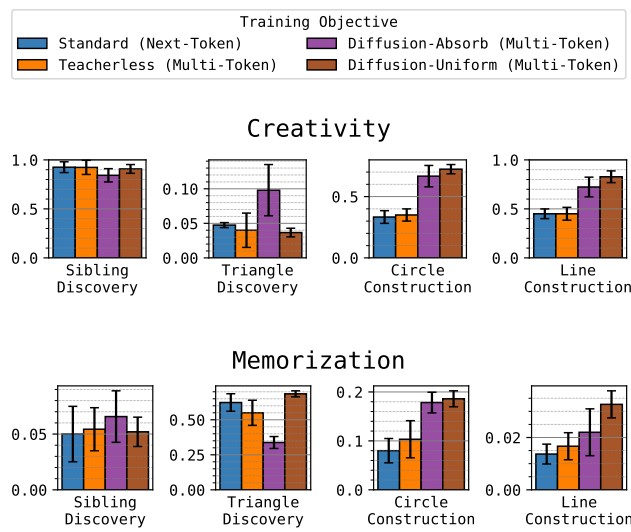

*Figure 4.* **Multi-token diffusion training improves algorithmic creativity (top; Eq 1) on three of our four open-ended tasks on `GPT-2 (86M)` and similarly-sized diffusion model, `SEDD (90M)`.** We report the best performance after tuning the sampling hyperparameters – temperature from $\{0, 0.5, 1, 2\}$, with or without top-$K$ where $K = 50$.

the smaller models, in Fig 4, we find that the diffusion model improves over next-token training of similar-sized Transformers on all tasks, except on `Sibling Discovery` where it is mildly worse. With teacherless training, the gains are absent; however we find that the creativity of the top-$K$ samples is generally improved (Fig 24), indicating a less peaky distribution. Regardless, the overall trend echoes prior findings that for smaller Transformers, teacherless training is hard to optimize and other multi-token objectives even hurt (Gloeckle et al., 2024) smaller models.

**Multi-token prediction reduces memorization for the larger model.** Algorithmic creativity may suffer either due to either incoherence or mode collapse or memorization. For our larger model, it is the last reason that dominates: across the board (in Fig 3), next-token prediction is significantly prone to memorizing the data, while the multi-token method is highly resistant. As foreshadowed in §2.6, we hypothesize that this is because NTP memorizes the earlier training tokens without a global plan, having fit the later tokens via local coherence rules (because of Clever Hans cheats à la B&N'24). In contrast, the smaller models have much lower absolute memorization values; the multi-token objective preserves or mildly worsens this, in a way that does not always hurt creativity proportionally. here, we find that amongst the top-$K$ samples, teacherless training has reduced memorization for some tasks (see Fig 25).

We point the reader to §C.6 for further empirical evidence supporting our argument about NTP from §2.6, including

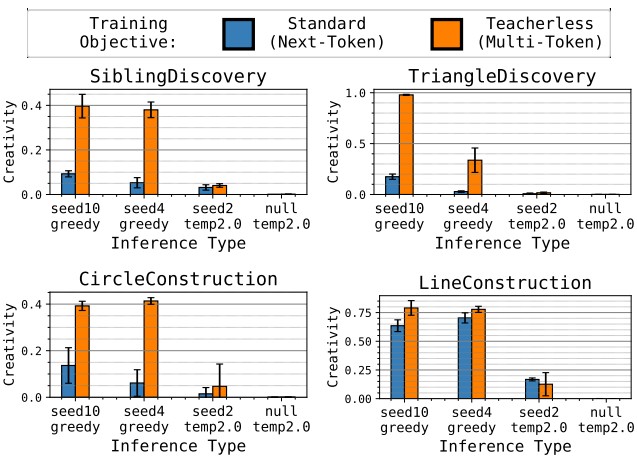

Figure 5. **Seed-conditioning significantly improves algorithmic creativity of both next- and multi-token prediction on Gemma v1 (2B) model.** The X-axis labels denote the prefix (at training and inference) and the temperature (at inference).

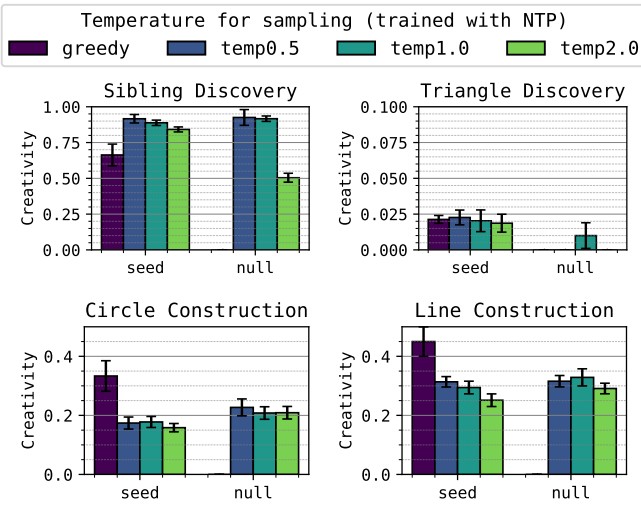

Figure 6. **Seed-conditioning achieves comparable algorithmic creativity in the GPT-2 (86M) model**: All models are trained with NTP. The color denotes temperature, while the X-axis, the prefix. Remarkably, seed-conditioning achieves comparable creativity even with greedy decoding, and improves creativity particularly in our exploratory creativity tasks (**bottom**).

experiments on token-reordering and experiments ruling out other hypotheses.

**Seed-conditioning improves algorithmic creativity for Transformers.** Orthogonal to the effect of multi-token vs. next-token objectives, we point out the effect of seed-conditioning a Transformer. First, and most remarkably, with seed-conditioning, there is no need for temperature: even deterministic greedy decoding generates diverse outputs and highly non-trivial creativity (Fig 5, Fig 6). This means that, even though the seeds we provide are arbitrarily paired with the training datapoint, the models have learned to transform the arbitrary noise into meaningful randomness. Next, increasing the seed lengths consistently boosts algorithmic creativity for both next-token and multi-token approaches (Fig 5, 23), presumably up to a limit. Thus, for Transformers, we propose viewing seed-conditioning as a knob for diversity distinct from temperature sampling.

Next, seed-conditioning results in algorithmic creativity comparable to temperature sampling. For the larger models (Fig 5), seed-conditioning generally outperforms the baseline "null" conditioning (which appears mode-collapsed, see Fig 18, 19, 20). For the smaller model as well (Fig 6), we find gains in all tasks except for Sibling Discovery. Besides, for any fixed temperature, prefixing a seed only improves performance over a null prefix (Fig 6, 18).

In §I.2, we find that the improved creativity from seed-conditioning comes from improved diversity, rather than from reduced memorization (unlike with multi-token training). All these behaviors of seed-conditioning require further study as they are non-trivial, especially given that the seeds are naively, arbitrarily chosen in relationship to the output. As a starting point, we provide some preliminary

hypotheses for when and why seed-conditioning may help Transformers in §C.1. Note that we do not see gains of seed-conditioning when it comes to diffusion training (§ G.3).

**Robustness to hyperparameters.** In §F and Fig 22, we do sensitivity analysis on all the datasets. We report how our above findings are robust to the choice of learning rate, batch-size, number of training steps, weight given to the multi-token objective, varying sampling conditions and reasonable changes to the complexity of the dataset and training set size (as per our argument in §2.6, we do expect the next-vs. multi-token gap to diminish for larger dataset size).

### 4.2. An exploration of real-world summarization

For a more realistic examination, we finetune GPT models NTP and the multi-token teacherless objectives on summarization tasks (XSUM, CNN/DailyMail). We measure the diversity of a model for any given prompt by generating 5 different completions and computing a Self-Bleu metric (Zhu et al., 2018). Admittedly though, a summarization task is not as open-ended as we would like: a higher quality model (i.e., higher Rouge; Lin, 2004) necessarily means lower diversity. To account for this, we plot how diversity evolves over time as a function of generation quality; we then find in Fig 7 that for a given model quality, the larger multi-token models achieve higher diversity (albeit only by a slight amount). This increase does not hold for smaller models and is not always noticeable for CNN/DailyMail (see §J). Interestingly, teacherless training consistently shows an increase in summarization quality, measured by Rouge.

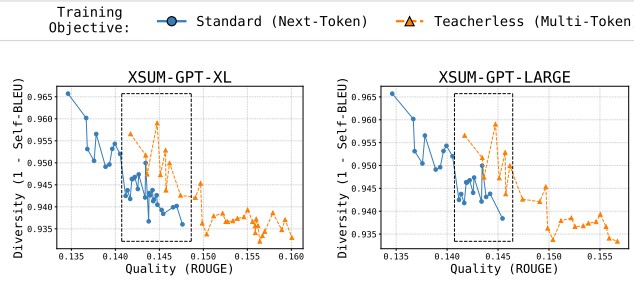

*Figure 7.* **Multi-token training improves diversity scores for XSUM summarization for large GPT-2 models**: We observe gaps in diversity for a fixed quality over the course of finetuning.

## 5. Related Work

**Open-ended algorithmic tasks.** Directly related to us are Khona et al. (2024); Allen-Zhu & Li (2023b) who study diversity of next-token-trained models on an open-ended algorithmic task. Khona et al. (2024) demonstrate a diversity-accuracy tradeoff under temperature-scaling for path-connectivity on on a knowledge graph. We show how this tradeoff can be improved under alternative training methods. Allen-Zhu & Li (2023b) empirically demonstrate that next-token predictors *are* able to learn a synthetic, challenging CFG, in the "infinite" data regime ($\approx 100m$ tokens). Our datasets are not CFGs, with the exception of `Sibling Discovery`. Our negative result does not contradict theirs since what we show is a sub-optimality of NTP in a smaller data regime. Our work also extends the above works by studying limitations in much more minimal tasks that require as little as 2-hop lookahead. There are other works that study Transformers on non-open-ended graph-algorithmic tasks, discussed in §K.

**Diversity in generative models.** Generative diversity has long been a major goal, at least until the revolution in (deterministic) reasoning of language models. Much work has gone into concerns such as mode collapse (Che et al., 2017) or posterior collapse (Bowman et al., 2016) and memorization (Carlini et al., 2020; 2023; Nasr et al., 2023). Temperature sampling has been known to have weak correlation with creativity, often inadvertently introducing incoherence (Peeperkorn et al., 2024; Chen & Ding, 2023; Chung et al., 2023). Our results on seed-conditioning are also reminiscent of a line of work on reinforcement learning (RL) showing that adding noises to the policy model parameters enables more efficient exploration than directly adding noises to the output space (Plappert et al., 2017; Fortunato et al., 2017). We defer discussion of theoretical studies of diversity and memorization and with empirical studies on natural-language creativity in §K.

**Going beyond next-token prediction (NTP).** There has been a recent emerging discussion surrounding the role of NTP as foundational piece in developing intelligent models. On the critical side, arguments have been made about the inference-time issues with auto-regression (Dziri et al., 2024; LeCun, 2024; Kääriäinen, 2006; Ross & Bagnell, 2010). Others have reported the planning and arithmetic limitations of next-token trained models (McCoy et al., 2023; Momennejad et al., 2023; Valmeekam et al., 2023a;b;c; Bachmann & Nagarajan, 2024) where the goal is accuracy, not diversity. As for diffusion as an alternative to NTP, our findings parallel that of Ye et al. (2024) who show that their variant of diffusion is able to solve the challenging path-star task of B&N'24. We provide references to more lines of multi-token prediction work in §K. There are also other Transformer failures such as the reversal curse (Allen-Zhu & Li, 2023a) or shortcut-learning (Dziri et al., 2024; Zhang et al., 2023; Liu et al., 2023; Young & You, 2022; Lai et al., 2021; Ranaldi & Zanzotto, 2023), however these are out-of-distribution failures; the sub-optimality we show is in-distribution, like in B&N'24.

**Injecting noise into a Transformer.** Most related to seed-conditioning is DeSalvo et al. (2024) who induce diversity by varying a *soft*-prompt learned using a reconstruction loss. Our approach requires no modification to the architecture or the loss; however, we train the whole model, which is more expensive. A concurrent position paper (Jahrens & Martinetz, 2025) conceptually suggests injecting noise with the same motivation as us. Seed-conditioning may also be viewed as a controllable way to vary prompt wordings which is known to induce diverse outputs (Li et al., 2023; Lau et al., 2024; Naik et al., 2024; Li et al., 2022).

## 6. Conclusions

This work provides a minimal test-bed of tasks abstracting distinct modes of creativity. While these tasks are admittedly an extreme caricaturization of real-world tasks, they enable us to quantify otherwise elusive metrics like originality and diversity. They also enable us to control and investigate distinct parts of the current apparatus for language modeling (next-token learning and temperature sampling) and advocate for alternatives (multi-token learning and seed-conditioning). Next, one can investigate the effect of length generalization, scaling and the effect of in-context learning vs. finetuning. The surprising effectiveness of seed-conditioning raises various open questions (§C.3). Other profound questions arise, such as whether reasoning-enhancing methods like RL and CoT are optimal for enhancing open-ended diversity and originality (§C.2). We hope our work inspires discussion in the various directions of multi-token prediction, creativity and planning. We present more discussions in §C and limitations in §A.

## Impact Statement

This paper presents work whose goal is to advance the field of Machine Learning through the study of simple algorithmic tasks inspired by creativity. There are many potential societal consequences of our work — especially if one applies AI to real-world creative endeavors — none which we feel must be specifically highlighted in our focused algorithmic study.

## Acknowledgements

We wish to thank Gregor Bachmann, Jacob Springer, and Sachin Goyal for extensive feedback on a draft of the paper. We are grateful to Vansh Bansal whose significant effort in an independent reproduction of our GPT-2 (86M) experiments helped zero in on the role of top-K sampling in seed-conditioning for some of our datasets which we had overlooked in initial versions of the paper. We wish to thank Quentin Fournier for excellent references to work on AI for scientific discovery. We thank Garret Tanzer and Elan Rosenfeld for helping arrive at the term "seed-conditioning". We also wish to thank Mike Mozer, Suhas Kotha, Clayton Sanford, Christina Baek, Yuxiao Qu, Ziqian Zhong for valuable discussions and pointers. The work was supported in part by Cisco, Apple, Google, OpenAI, NSF, Okawa foundation and Schmidt Sciences.

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

# A. Limitations

We enumerate in detail the limitations of our work in terms of our experimental conclusions and in terms of our general approach to an abstraction of creativity.

## A.1. Limitations of our experimental conclusions

1. There may be many ways to improve upon next-token prediction for a minimal task. Unfortunately, success here does not necessarily guarantee success on more complex tasks. Conversely, minimal tasks are more valuable as a failure base case: failure here guarantees failure in more complex tasks.

2. Our examples do not preclude the existence of tasks where next-token prediction will outperform multi-token prediction; multi-token prediction is simply a more general-purpose objective suitable to lookahead tasks.

3. The teacherless multi-token prediction technique we explore as an alternative is generally harder to optimize than next-token prediction, especially for smaller models.

4. Even if multi-token approaches outperform next-token prediction relatively, in some of our simple tasks, all algorithms are far from delivering a sufficiently diverse model.

5. Although our tasks are minimal, we note that there is a certain range of hyperparameters (e.g., high degree or edge count) beyond which the models can struggle to learn them. We find that `Triangle Discovery` in particular is a challenging task, especially for smaller models. We also note that the models are curiously sensitive to the way the edges are formatted (see §G.5).

6. While seed-conditioning appears to be a competitive alternative to temperature sampling, there are caveats here:

   (a) In our `GPT-2 (86M)` experiments for `Sibling Discovery` and `Triangle Discovery`, the gap between seed-conditioning and temperature-sampling is contingent on top-K sampling (which is turned on by default in the library as it is standard practice in real-world applications). Although this does not seem to be a requirement in our other settings (and is immaterial for the other datasets which have a smaller vocabulary), it is not clear how crucial this factor would be for demonstrating the same gap in real settings.

   (b) It is not clear how well this idea may generalize to realistic data.

   (c) Unfortunately, seed-conditioning requires special training of the model, which is not as appealing as the far simpler temperature sampling.

7. There are many ways in which our observations have not been fully characterized or understood even in our minimal settings. For example, we do not thoroughly report on the effect of model size, model family, or pretraining; we do not analyze when and why seed-conditioning helps.

## A.2. Our approach to creativity

Below, we enumerate some important limitations of our approach towards building abstract and minimal models of creative tasks.

1. The skills we capture in our tasks are only (a subset of) computational skills *necessary* for creativity; these are far from being *sufficient*.

2. The type of algorithmic tasks we study capture only a tiny subset of creative tasks that fall under the taxonomy in Boden (2003). There is yet another class called *transformative* creativity that we do not look at, and also other important taxonomies such as the Big-C/little-c creativity (Csikszentmihalyi, 1996). Big-C Creativity corresponds breakthroughs and world-changing ideas; what we focus on is adjacent to a class of little-c creativity tasks. Relatedly, many real-world creative tasks appear to be "out-of-distribution" in nature, which we do not capture.

3. Real-world creative tasks also apply over much larger context length and require drawing connections from a significantly larger memory (literally, the set of all things a human may know about). Our algorithmic tasks are tiny in comparison (although deliberately so).

4. Our empirical measure of creativity for algorithmic tasks is only a computationally-efficient proxy. Achieving an absolute high algorithmic creativity score does not imply a complete coverage of the space.

5. As stated earlier, we study abstract tasks that are inspired by the computations involved in creative tasks. Our study is not intended to capture the subjective aspects like interestingness (Schmidhuber, 2009) and other social, cultural and personal values integral to many creative tasks.

## B. Transformer Training Objectives

Let $\text{LM}_\theta$ be our language model, parameterized by $\theta$, for which $\text{LM}_\theta(\hat{s}_i = s_i; \boldsymbol{s}_{<i})$ is the probability it assigns to the $i$th output $\hat{s}_i$ being $s_i$, given as input a sequence $\boldsymbol{s}_{<i}$. Let $(\boldsymbol{p}, \boldsymbol{r})$ be a prefix-response pair. In standard next-token finetuning, we maximize the objective:

$$\mathcal{J}_{\texttt{next-token}}(\theta) = \mathbb{E}_\mathcal{D}\Big[ \sum_{i=1}^{L_{\texttt{resp}}} \log \text{LM}_\theta\left(\hat{r}_i = r_i; \boldsymbol{p}, \boldsymbol{r}_{<i}\right) \Big] \tag{2}$$

In teacherless (multi-token) training (Monea et al., 2023; Bachmann & Nagarajan, 2024; Tschannen et al., 2023), we make use of an uninformative input string $\$$ that simply corresponds to a series of dummy tokens $\$$.

$$\mathcal{J}_{\texttt{multi-token}}(\theta) = \mathbb{E}_\mathcal{D}\Big[ \sum_{i=1}^{L_{\texttt{resp}}} \log \text{LM}_\theta\left(\hat{r}_i = r_i; \boldsymbol{p}, \boldsymbol{\$}_{<i}\right) \Big] \tag{3}$$

## C. Further discussion

### C.1. Intuition for seed-conditioning

Seed-conditioning produces non-trivial algorithmic creativity even with greedy decoding. Perhaps, one could view the seed prefixes as a simpler alternative to varying the wordings of a prompt (Li et al., 2023; Lau et al., 2024; Naik et al., 2024; Li et al., 2022) or tuning a soft-prompt (Wang et al., 2024c), both of which are known to induce diversity. However, it is unexpected that the seeds provide any meaningful randomness at all, given they are arbitrarily picked with no semantic relationship to the training point. We do not understand this presently.

Another pertinent question is to why seed-conditioning works better than temperature sampling in the settings it does. To this, we offer three potential explanations. The simplest explanation is that in the initial version of our paper, top-K sampling was enabled by default with $K = 50$ for the GPT-2 (86M) settings. This would have restricted the degree of freedom in the support induced by the output-layer randomness; thus, providing different seeds can help vary this support and produce more varying samples. However, even upon removing top-K sampling, we see gains in our exploratory creativity tasks for GPT-2 (86M) (after all their vocabulary is smaller than $K = 50$), and in all our Gemma v1 (2B) results. This is unexplained.

Thus, we speculatively put forth two more arguments. First, is a representational one. Fixing a random seed upfront may help the model flesh out (i.e., compute the tokens of) one thought per sample, as against maintaining a running set of multiple thoughts and computing a distribution over all their tokens at each step. A similar point is made in a concurrent position paper (Jahrens & Martinetz, 2025). The second argument is specific to next-token prediction on open-ended planning tasks: fixing a random seed upfront may help the model co-ordinate multiple interlocking random decisions in advance rather than deciding them on the fly.

Finally, there are also optimization aspects of how seed-conditioning works that we do not understand (see §C.3). Regardless, it remains to be seen whether seed-conditioning is useful in tasks beyond the minimal ones we design.

### C.2. Effects of reasoning-enhancing methods.

Our argument is limited to learning open-ended tasks in a supervised manner which directly corresponds to how we perform pretraining or apply models for protein or drug discovery. While we do not comment on how well other powerful approaches like RL (DeepSeek-AI, 2025), chain-of-thought (CoT; Wei et al., 2022), and scaling test-time compute (OpenAI, 2024) would address the limitations we bring up, we make a few remarks. First, in cases where post-training only elicits pre-existing skills in the base model (Muennighoff et al., 2025), our limitations and suggestions about a next-token-trained base model remain relevant. Next, regardless of the benefits one may get from pre-training, our arguments imply that

pre-training squanders its rich supervision, when it could be vastly compute- and data-efficient. Finally, there is a profound question as to whether exploration through sparse rewards can learn the creative process during training-time or whether articulation in the token space (with a thinking model) can efficiently maximize diversity during test-time (which may require brute-force enumeration of all candidates).

### C.3. Style of noise-injection

Our technique of injecting noise into the model is somewhat different from how noise is introduced in traditional VAEs (Kingma & Welling, 2014) or GANs (Goodfellow et al., 2020), and this difference is worth noticing. In traditional approaches, although the model learns a noise-output mapping, this mapping is enforced only at a distribution level i.e., the distribution of noise vectors must map to a distribution of real vectors. However, in our approach we arbitrarily enforce what noise vector goes to what real datapoint, at a pointwise level. This raises the open questions of why seed-conditioning works in the first place — surprisingly, without breaking optimization or generalization — and whether there is a way to enforce it at distribution-level, and whether that can provide even greater improvements.

### C.4. In-weights vs in-context graphs for combinational creativity

Combinational creativity requires searching through known entities. In abstracting this, there is an interesting choice to be made as to whether the relevant search space is retrieved and spelled out in-context or whether it remains in-weights (like in `Sibling Discovery` and `Triangle Discovery`). We argue that the in-context version does not capture the creative skills required in many real-world tasks. For instance, discovering a fresh and surprising analogy necessitates noticing similarities from sufficiently distinct parts of one's vast, rich space of memory. Thus, the core challenge here lies in retrieving from the entirety of one's memory. If one were to faithfully simulate this an in-context version of this in a model, one would have to provide the entirety of the model's pretraining data in context.

### C.5. Examples of `Triangle Discovery`

Although we presented this task as a more complex, higher-order counterpart to `Sibling Discovery`, we retrospectively identify some real-world examples that resemble the higher-order search skill involved in this task.

1. **Discovering contradictions:** Consider identifying non-trivial contradictions within (a large body) of knowledge (like a legal system, or a proof based on many lemmas, or the literature spanning many papers in a certain field). This may require identifying two or more facts that together result in an implication that contradicts another fact.

2. **Discovering feedback loops:** Fields like biology, ecology, climate science and economics may involve discovering non-trivial feedback loops. Unlike feedback loops where two events encourage each other, a non-trivial loop would be one where an `Event A` encourages `Event B`, that in turn encourages `Event C` that in turn encourages `Event A`.

3. **Antanaclasis:** An antanaclasis involves using a word in two different senses in a sentence, while still ensuring that each sense has a coherent relationship with the rest of the sentence. Consider Benjamin Franklin's quote, `Your argument is sound, nothing but sound`. Here, the two senses are $sound_1$ as in "logically correct", $sound_2$ as in "noise". This sentence encodes an pairwise relationship between three entities $\{argument, sound_1, sound_2\}$ individually. While the last two entities (the two senses) themselves must be related to each other (through the common word, `sound`), for a coherent sentence, both senses must also be appropriate descriptors for the first entity, `argument`. Thus, constructing this sentence requires searching through one's vocabulary to discover three words that satisfy these three relationships simultaneously.

4. **Word games:** Some word games require identifying a set of words that simultaneously have pairwise relationships with each other.

   (a) For example, standard crosswords would require identifying sets of 4 or more words that have various simultaneous pairwise intersections in the letters used.

   (b) Devising "& Lit." clues in cryptic crosswords are an altogether different, yet compelling example that require discovering a satisfying triangular relationship. Consider the clue "`Some assassin in Japan`" whose answer is `Ninja`. Here the phrase `Some assassin in Japan` participates in two senses. First, is the direct semantic sense as a definition of what a Ninja is. But there is a second, indirect

sense: the word `Some` indicates that the solution lies as some substring of the phrase, namely `"as-sassi(n in Ja)pan"`. Thus, constructing the clue requires identifying a triangular relationship between $\{\text{Ninja}, (\text{Some assassin in Japan})_1, (\text{Some assassin in Japan})_2\}$ just like in an antanaclasis. This is true generally of any & Lit. clues as these clues must perform "double duty" in pointing to the answer.

### C.6. Further evidence of our argument in §2.6

Below we provide two more pieces of evidence affirming the failure mechanism of next-token prediction outlined in §2.6.

**Improved algorithmic creativity is not due to some form of capacity control.** While §2.6 argues that multi-token prediction should help creativity by providing critical lookahead capabilities, it is also possible that it simply acts as a form of capacity control that prevents memorization. We rule this out in Fig 8: even as memorization computed on *unseen* seeds is controlled, the multi-token model perfectly reproduces the training data on *seen* seeds. We term this *seed-memorization*. An exact equivalence of this phenomenon was noticed in GANs in Nagarajan et al. (2018), where the generator can be trained on specific latent vectors to memorize the mapping on those, and yet produce fresh samples outside of those latent vectors.

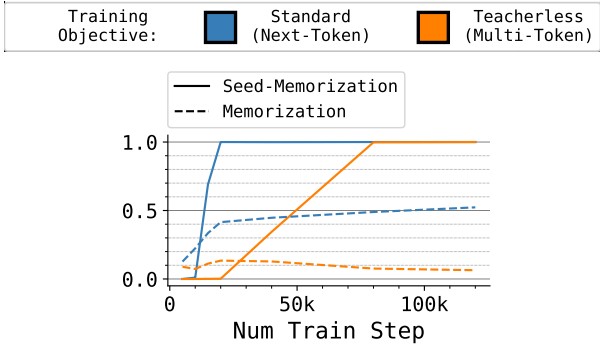

*Figure 8.* **Even if multi-token prediction reduces memorization (on unseen seeds), it has enough capacity to memorize training data on the seen seeds (denoted by seed-memorization).** Note that the best algorithmic creativity for `NTP` and `MTP` are achieved at step 10k and 40k, respectively, which are the checkpoints we used to report metrics in Fig 4.

**Effect of token reordering.** The implication of our argument in §2.6 is that next-token learning would benefit from reversing the token ordering of the `Sibling Discovery` task (i.e., parent appears before siblings). Indeed, we find this to be the case in Fig 12 and Fig 22. Interestingly, we find that the reverse-trained `NTP` model is still far from the original multi-token teacherless model. More surprisingly, a teacherless model trained on the reversed data, achieves even higher algorithmic creativity of all training methods here. Note that in all other datasets, no reordering of the tokens should make any change to the training.

## D. Description of datasets

### D.1. Datasets inspired by combinational creativity

**Dataset 1: `Sibling Discovery` (Fig 9(a)).** This task is based off a bipartite graph $\mathcal{G}$ made of parent vertices $\mathcal{V} = \{A, B, C, \ldots\}$ each neighboring a corresponding set of children $\text{nbr}(A) = \{a_1, a_2, \ldots, \}$. We set the number of parent vertices $|\mathcal{V}|$ to be small and the number of children for each parent vertex $|\text{nbr}(A)|$ to be large. For example, $|\mathcal{V}| = 5$ and $|\text{nbr}(A)| = 500$. We define $\text{coh}(s)$ to hold on "sibling-parent" triplets of the form $s = (\gamma, \gamma', \Gamma)$ such that $\gamma, \gamma' \in \text{nbr}(\Gamma)$.

Next, we ensure that the training set is large enough for the model to infer all the edges in the graph. Let $m = |\mathcal{V}|$ and $n = |\text{nbr}(\Gamma)|$ (for all $\Gamma \in \mathcal{V}$). This means $S = \Omega(m \cdot n)$. At the same time, to keep the task non-trivial, the training set must be small enough to not cover all the coherent sibling-parent triplets. Thus, we ensure $S = o(m \cdot n^2)$.

**For the default version of this dataset,** we set $|\mathcal{V}| = 5$ and $|\text{nbr}(\Gamma)| = 500$ for all $\Gamma \in \mathcal{V}$.

**Dataset 2: `Triangle Discovery` (Fig 9(b)).** This task is based off an undirected graph $\mathcal{G} = (V, E)$ which contains many triangles. Since a triangle is a symmetric structure, the problem remains the same even upon reordering the vertices. Thus, in this task $\text{coh}((v_1, v_2, v_3)) = \text{true}$ iff all three edges between $\{v_1, v_2, v_3\}$ belong in $\mathcal{G}$. To make this task interesting

**(in-weights graph)**

**Training data**

(1) **"g, f, Y"**
(2) **"e, d, Y"**
(3) **"h, c, X"**
(4) **"b, i, X"**

**Generated samples**

(A) **"b, f, Y"** (incoherent)
(B) **"b, i, X"** (memorized ~ (4))
(C) **"e, d, Y"** (memorized ~ (2))
(D) **"e, d, X"** (incoherent)
(E) **"X, i, b"** (incoherent)
(F) **"b, c, X"** ✓
(G) **"e, g, Y"** ✓
(H) **"e, g, Y"** (duplicated ~ (G))

**Algorithmic creativity = 2 / 8**

(a) Sibling Discovery

**(in-weights graph)**

**Training data**

(1) **"tri: ab, bc, ca"**
(2) **"edge: cd, dc"**
(3) **"edge: ab, ba"**
(4) **"edge: bc, cb"**
(5) **"edge: fb, bf"**
(6) **"edge: ac, ca"**
(7) **"tri: da, ac, cd"**
(8) **"edge: ce, ec"**
(9) **"edge: ed, de"**
(10) **"edge: ad, da"**

**Generated samples**

(A) **"tri: ab, bc, ca"** (memorized ~ (1))
(B) **"tri: ab, bf, fa"** (incoherent)
(C) **"tri: bc, ca, ab"** (memorized ~ (1))
(D) **"tri: ca, ab, bc"** (memorized ~ (1))
(E) **"tri: cd, de, ea"** ✓
(F) **"tri: de, ea, ad"** (duplicated ~ (E))
(G) **"tri: dc, ce, ed"** (duplicated ~ (E))
(H) **"tri: bc, ce, eb"** (incoherent)
(I) **"tri: dc, ca, ad"** (memorized ~ (7))
(J) **"tri: ac, ce, ea"** (incoherent)

**Algorithmic creativity = 1 / 10**

(b) Triangle Discovery

*Figure 9.* **Minimal tasks inspired by combinational creativity:** The in-weights graph represents the underlying knowledge graph used to generate the training data (not provided in-context). Based on our definition of algorithmic creativity in Eq. (1), generated samples that are incoherent or memorized, or duplicated are not counted as valid samples. Note that sequences that are permutations of each other are considered identical when computing duplicates and memorization.

(neither too trivial nor too non-trivial) for our models to learn, we enforce several constraints on the graph. First, we try to keep the degree deg of each vertex to be sufficiently small. On the one hand, this is so that no vertex requires too much computation to find a triangle it is part of; on the other, we also do not want a very dense graph where most random triplets are a triangle. In addition to this degree requirement, we ensure that each vertex has a minimum number of triangles.

Thus to create a graph that is neither too trivial nor too non-trivial, we define a two-step graph generation procedure. In the first step, we iterate over the vertices, and add deg many edges from that vertex to other vertices in the set (where deg is small, such as 3 or 10). To avoid creating high-degree vertices inadvertently, we only select neighbors with degree $\leq 1.2 \cdot$ deg. This alone may not ensure a sufficient number triangles in each vertex; so we iterate over the vertices to explicitly create tri random triangles on each vertex (where tri is small, such as 6 or 10). We do this by selecting pairs of a vertex's neighbors and drawing an edge between them.

Next, we want a training dataset such that (a) the model can infer all the edges from the graph and yet (b) not all triangles appear in the dataset. This necessitates training on a dataset that consists not only of a subset of the triangles, but also of edges from the graph. Our training data consists of two parts: (1) $1/3$ are random triangles from the graph, (2) $2/3$ are random edges from the graph. In the training set, the triangle and edge samples are distinguished by a prefix "triangle:" or "edge:". During test-time, we ensure that the model is prompted with "triangle:". A triangle $(u, v, w)$ is tokenized as "tri: $(u, v), (v, w), (w, u)$" and an edge $(u, v)$ as "edge: $(u, v), (v, u)$". We provide both the directions of edge to potentially avoid any issues with the reversal curse (Berglund et al., 2024; Allen-Zhu & Li, 2023a).

**For the default setting of the dataset,** we set $|V| = 999$, deg $= 3$, tri $= 6$.

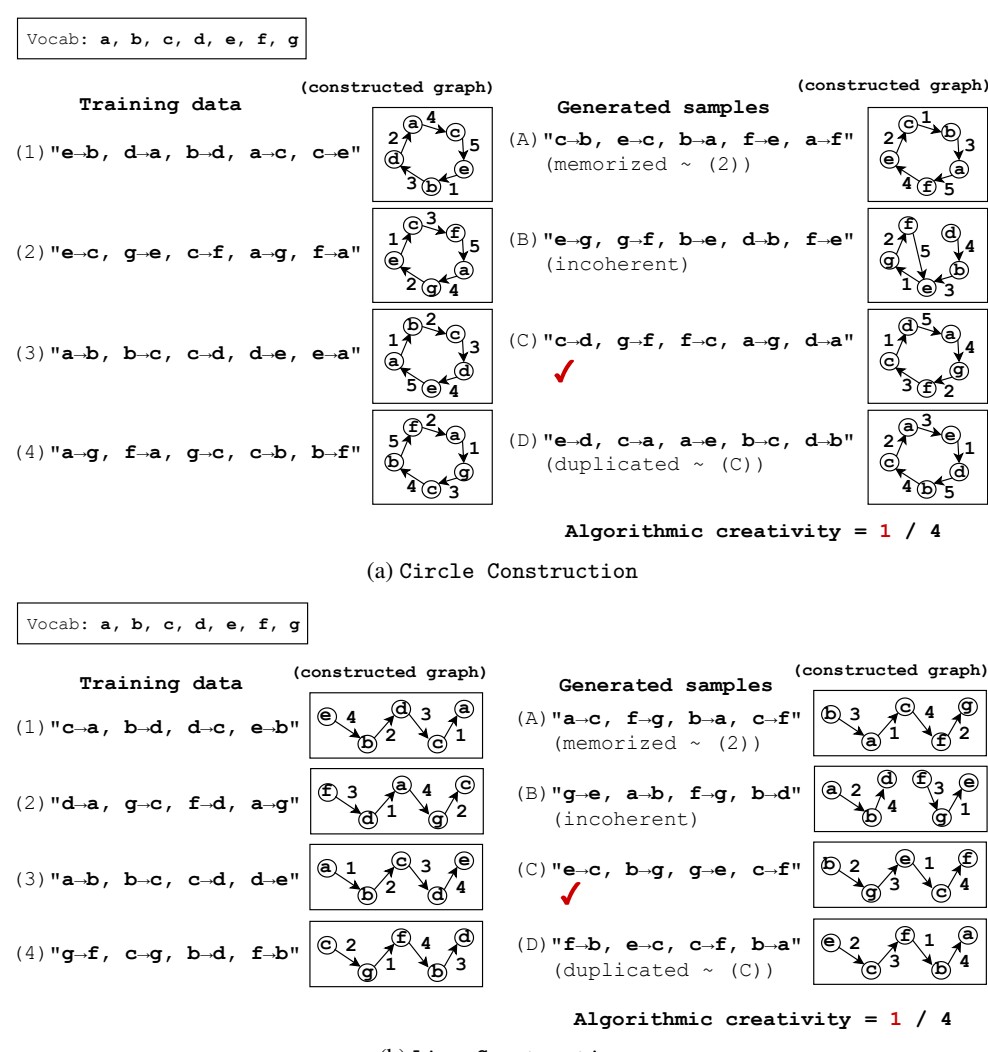

(a) Circle Construction

(b) Line Construction

*Figure 10.* **Tasks inspired by exploratory creativity:** The constructed graph visualizes the graph induced by the training or generated sample. Edge indices represent the order of edge appearing in the string. Based on our definition of algorithmic creativity in Eq. (1), generated samples that are incoherent, memorized, or duplicated are not counted as valid samples. Note that sequences that correspond to the same permutations but with different participating vertices are considered identical when computing duplicates and memorization

### D.2. Datasets inspired by exploratory creativity

**Dataset 3: `Circle Construction` (Fig 10(a)).** In this task, the generated strings must be randomized adjacency lists that can be rearranged to recover circle graphs of $N$ vertices. The vertices come from a fixed vocabulary of $M$ tokens. Specifically, let the generated list be $s = (v_{i_1}, v_{i_2}), (v_{i_3}, v_{i_4}), \ldots$. We define $\mathrm{coh}(s) = \mathrm{true}$ iff there exists a *resolving* permutation $\pi$ such that $\pi(s) = (v_{j_1}, v_{j_2}), (v_{j_2}, v_{j_3}), \ldots (v_{j_n}, v_{j_1})$ for distinct $j_1, j_2, \ldots j_n$. i.e., each edge leads to the next, and eventually circles back to the first vertex. In our experiments, we set $M$ to be larger than $N$.

**Our default experiments** are reported for $N = 9, M = 15$.

**Dataset 4: `Line Construction` (Fig 10(a)).** This task is a minor variant of the above where the edge set $E$ corresponds to a line graph. The details are same here except for coherence to hold, we need a resolving permutation $\pi$ such that $\pi(s) = (v_{j_1}, v_{j_2}), (v_{j_2}, v_{j_3}) \ldots, (v_{j_{n-1}}, v_{j_n})$ for distinct $j_1, j_2, \ldots j_n$. i.e., each edge leads to the next, stopping at a dead-end. We use the same set of hyperparamters as `Circle Construction`.

**Our default experiments** are reported for $N = 9, M = 15$.

*Table 1.* **Hyperparameter details for `Gemma v1 (2B)` model.**

| Hyperparameter | Sibling Discovery | Triangle Discovery | Circle Construction | Line Construction |
|---|---|---|---|---|
| Max. Learning Rate | $5 \times 10^{-4}$ | $5 \times 10^{-4}$ | $5 \times 10^{-4}$ | $5 \times 10^{-5}$ |
| Model Seq. Len. | 32 | 32 | 2048 | 2048 |
| Training steps | 7500 | $10k$ | $15k$ | $15k$ |
| Training size | $50k$ | $15k$ | $10k$ | $10k$ |
| Weight given to multi-token obj. | 0.5 | 0.5 | 0.75 | 0.75 |

*Table 2.* **Hyperparameter details for `GPT-2 (86M)` and `SEDD (90M)` model.**

| Hyperparameter | Sibling Discovery | Triangle Discovery | Circle Construction | Line Construction |
|---|---|---|---|---|
| Max. Learning Rate | $1 \times 10^{-4}$ | $1 \times 10^{-4}$ | $1 \times 10^{-4}$ | $1 \times 10^{-4}$ |
| Model Seq. Len. | 128 | 128 | 128 | 128 |
| Max training steps (`GPT-2 (86M)`) | $20k$ | $80k$ | $40k$ | $40k$ |
| Max training steps (`SEDD (90M)`) | $80k$ | $80k$ | $80k$ | $80k$ |
| Training size | $50k$ | $15k$ | $10k$ | $10k$ |
| Weight given to multi-token obj. | 0.5 | 0.33 | 0.75 | 0.75 |

# E. Further experimental details

**Details for `Gemma v1 (2B)` model.** In Table 1, we provide the hyperparameter details for each of our datasets. We note some common details here. First, the batch size is 4, but each sequence is packed with multiple examples; thus the model sequence length (divided by the input length) can be treated as a multiplicative factor that determines the effective batch size. The learning rates are chosen favorable to next-token prediction (not multi-token prediction). The training steps were chosen roughly based on a point after which the model had saturated in algorithmic creativity (and exhibited decreasing creativity). We use a learning rate with linear warm up for 100 steps, followed by cosine annealing upto a factor $0.01\times$ of the maximum learning rate. To measure creativity, we sample a test dataset $T$ of 1024 datapoints.

We represent the main tokens in our tasks with integers (ranging upwards of 0 to as many distinct integers as required). In the seed-conditioning setting, we use seeds of default length 10, using randomly sampled uppercase characters from the English alphabet. In all datasets, we space-separate the vertices in a string, and comma-separate the edges.

**Details for `GPT-2 (86M)` model.** In Table 2, we provide the hyperparameter details for each of our datasets. We use `GPT-2 (small)` with 86M non-embedding parameters when we are comparing Transformers with diffusion models. We train these models with a learning rate of $10^{-4}$ and a batch size of 64, to convergence in terms of the algorithmic creativity. In temperature sampling, we use nucleus sampling with a parameter of 0.95, although we observed this does not affect the results qualitatively. We used top-k sampling with a parameter of 50. We also explore temperatures above 1.0 for these models since temperatures low than 1.0 only did worse. We provide sensitivity analysis of learning rate in §G.

**Details for `SEDD (90M)` model.** We use SEDD's "absorb" variant, which begins denoising with a fully masked sequence and iteratively refines tokens over 128 denoising steps. This variant achieves the best language modeling performance in the original paper. Same as `GPT-2 (86M)`, we train these models with a learning rate of $10^{-4}$ and a batch size of 64, to

convergence in terms of algorithmic creativity. [1] We provide sensitive analysis of learning rate in §G. For sampling, we used the default function parameters, which turned out to involve top-$K$ sampling with $K = 50$.

---

[1]We use the codebase of Lou et al. (2023) at `https://github.com/louaaron/Score-Entropy-Discrete-Diffusion`.

## F. Sensitivity analyses for `Gemma v1 (2B)`

In this section, we report that our observations are robust to the choice of various hyper-parameters. First, we present a series of plots for the `Gemma v1 (2B)` model; each group of plots reports varying one hyperparameter for all the datasets. Fig 11 for train set size, Fig 12 for task complexity, Fig 13 for the weight given to the multi-token objective (and Fig 14 correspondingly for memorization), Fig 15 for learning rates, Fig 16 for number of training steps and Fig 17 for batch size. In §F.1, we report analyses for varying sampling conditions. It is worth noting that the occasional exceptions to our trends generally come from `Line Construction`, suggesting that this task is most friendly towards next-token prediction of the four we study.

**Note on task-complexity.** In Fig 12, we report robustness of our results to variations in the task complexity (e.g., degree, path length etc.,). Note that the variations we have explored are within reasonable factors. If we vastly increase certain factors (e.g., increase the degree of the vertices), we expect learning to become either highly trivial or non-trivial (see §D for some reasoning). Besides, as discussed in the main paper, teacherless training is a hard objective to optimize especially for smaller models; thus, we expect increasing the task complexity beyond a point to hurt the teacherless model for a fixed model size (crucially, for optimization reasons, not generalization reasons).

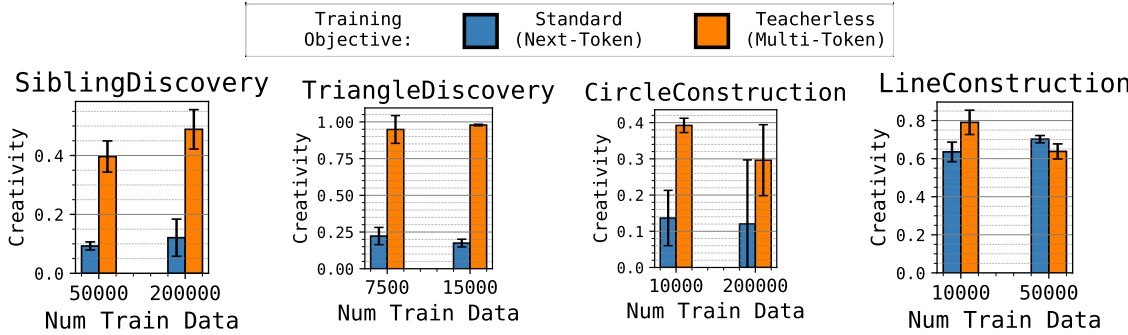

*Figure 11.* **Training size and algorithmic creativity for `Gemma v1 (2B)`:** Algorithmic creativity increases under multi-token prediction across various training set sizes. Note though that, in our examples, we except the gap to diminish eventually with sufficiently many training datapoints (this is unlike the failure of next-token prediction in B&N'24).

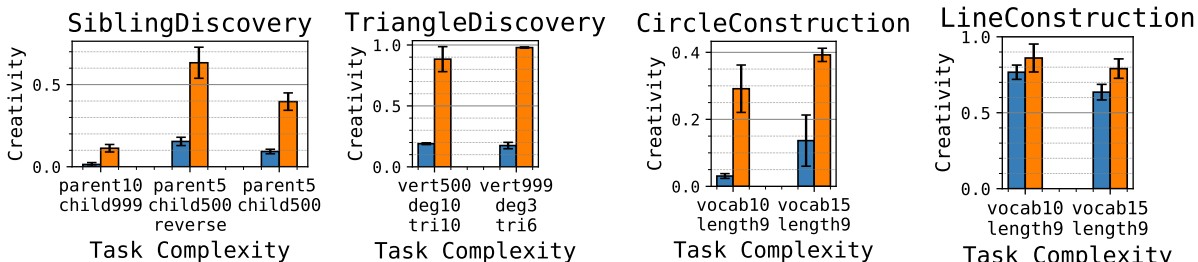

*Figure 12.* **Task complexity and algorithmic creativity for `Gemma v1 (2B)`:** Algorithmic creativity increases under multi-token prediction across (reasonable) variations in the dataset parameters (as described in §D).

### F.1. Varying sampling methods

Fig 18, Fig 19, and Fig 20 report creativity, memorization and coherence (i.e., fraction of generated strings that are coherent) for various sampling methods (greedy decoding and nucleus sampling) with various prefix conditionings. Here we not only look at the baseline (denoted by `null`) and seed prefix (denoted by `seed`), we also look at using a prefix of pause tokens (Goyal et al., 2024) (denoted by `pause`). This can be viewed as way to provide extra computation to the model before it emits its outputs. If this had the same effect of seed-conditioning, then we would conclude that the gains of seed-conditioning come purely from prepending extra computation to the model — not from the act of injecting noise at the input. However, in our experiments, we do not find noticeable gains from pause-conditioning, establishing that noise-injection is in itself beneficial.

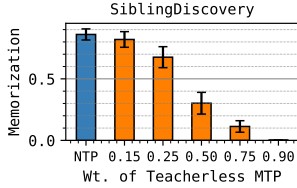 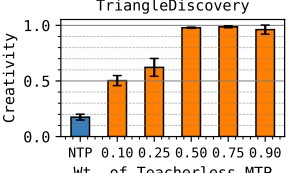 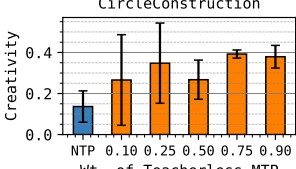 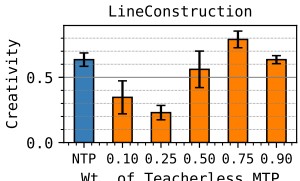

*Figure 13.* **Weight given to multi-token objective and algorithmic creativity for `Gemma v1 (2B)`:** Algorithmic creativity increases under multi-token prediction across various weights given to the multi-token component of the objective, barring some deviations for `Line Construction`.

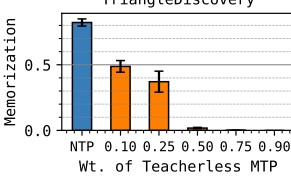 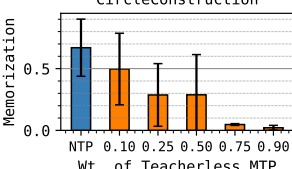 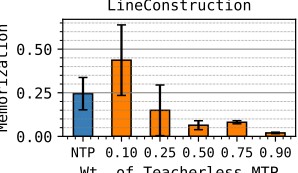

*Figure 14.* **Weight given to multi-token objective and memorization score for `Gemma v1 (2B)`:** Memorization reduces under multi-token prediction across various weights given to the multi-token component of the objective.

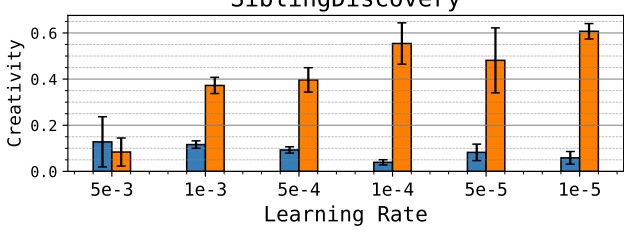 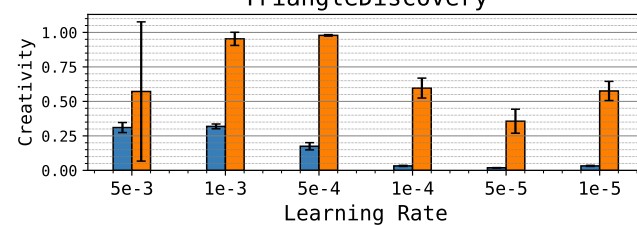

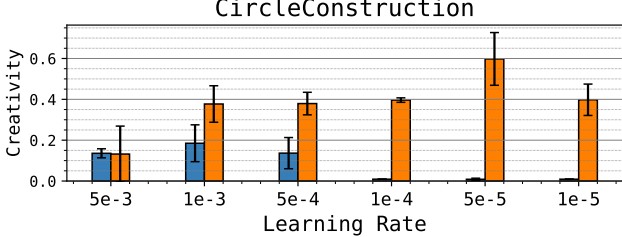 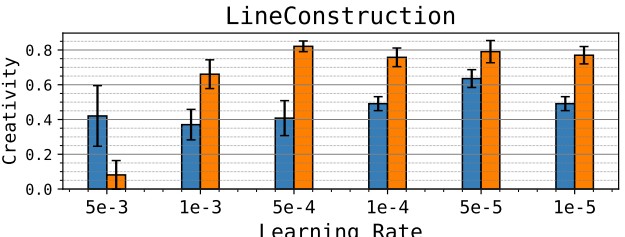

*Figure 15.* **Learning rate and algorithmic creativity for `Gemma v1 (2B)`:** Algorithmic creativity increases under multi-token prediction across various learning rates.

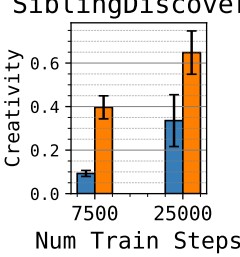 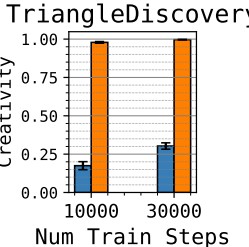 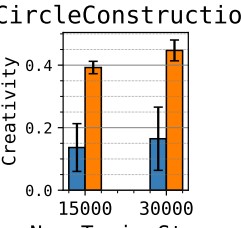 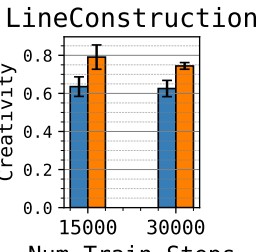

*Figure 16.* **Training steps and algorithmic creativity for `Gemma v1 (2B)`:** Algorithmic creativity under multi-token prediction across lengths of training.

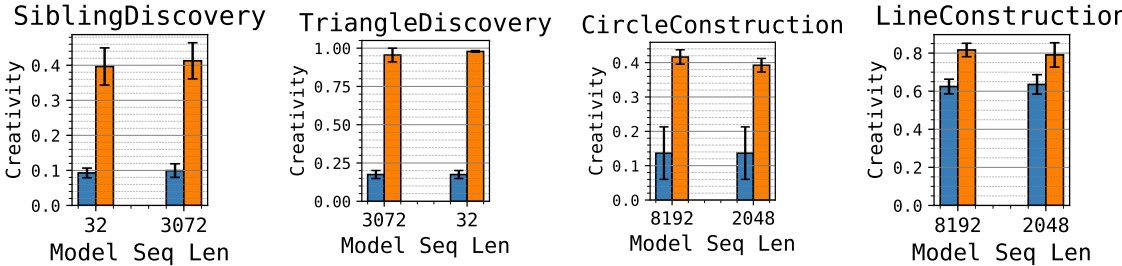

*Figure 17.* **Batch size and algorithmic creativity for `Gemma v1 (2B)`:** Algorithmic creativity increases under multi-token prediction across various batch sizes. Note that here batch size is effectively proportional to the model sequence length, since we pack multiple finetuning examples into the sequence.

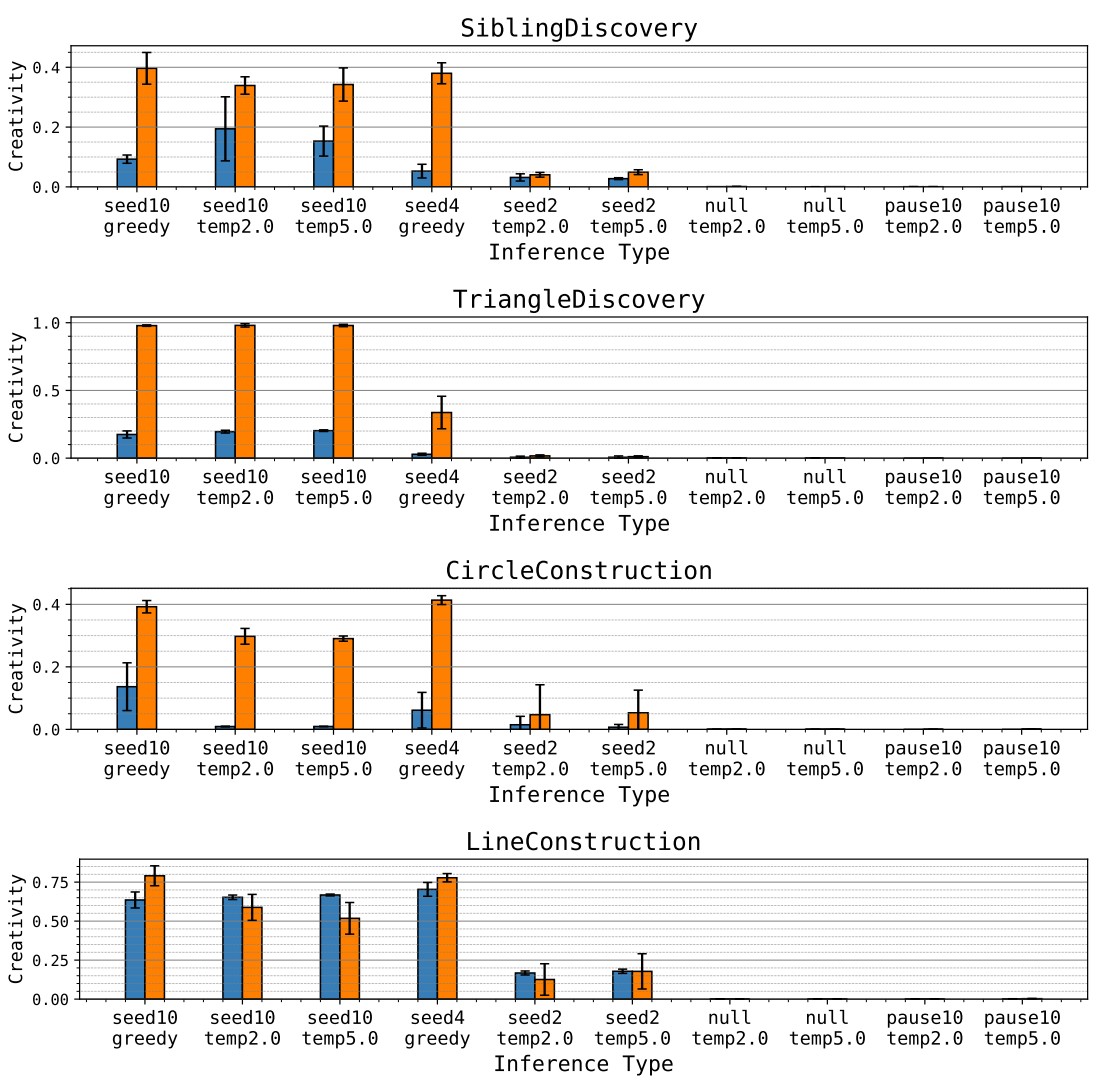

*Figure 18.* **Algorithmic creativity under various sampling conditions for `Gemma v1 (2B)`:** Across all conditions, and in almost all datasets (with a few exceptions in `Line Construction`), multi-token prediction improves creativity. Furthermore, seed-conditioning achieves best algorithmic creativity, with a longer seed helping more.

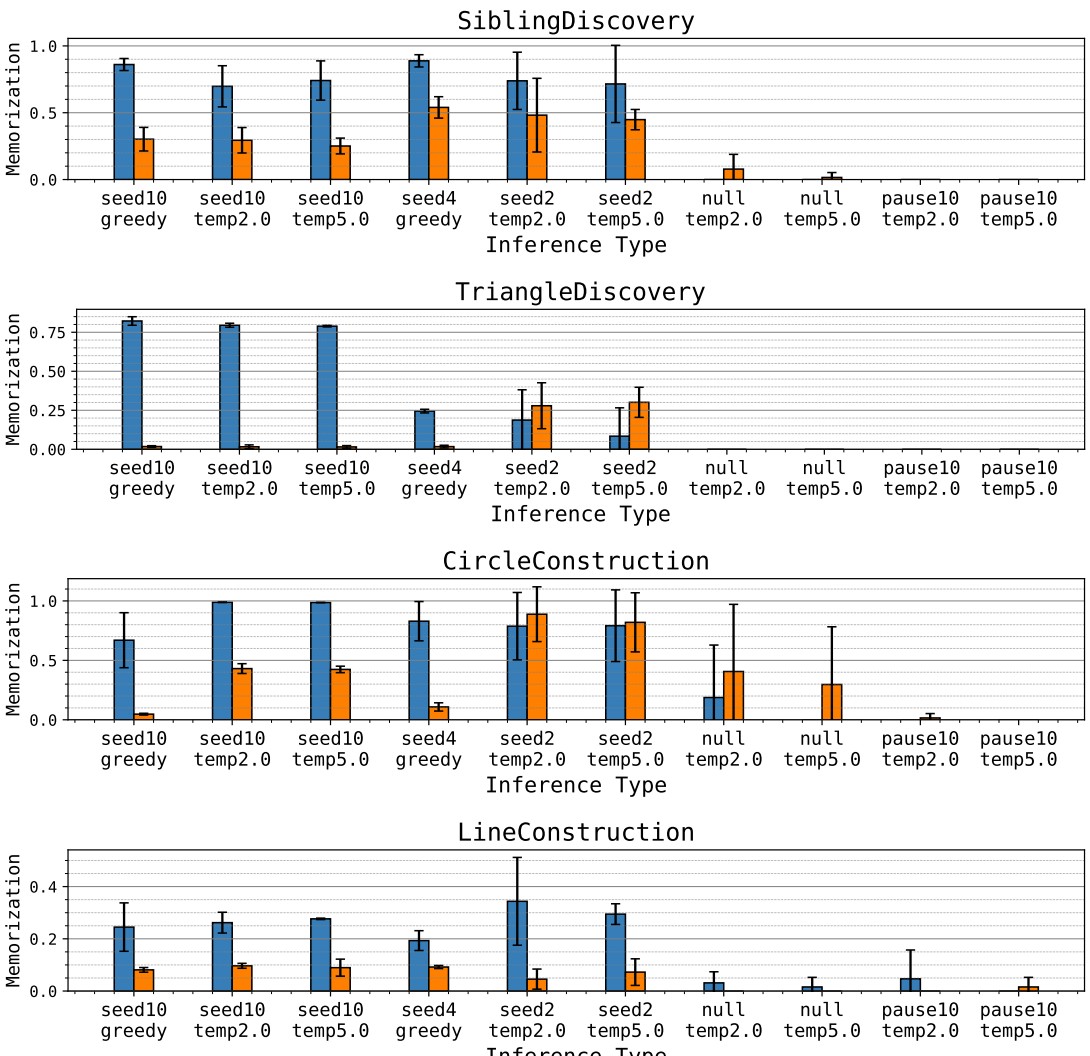

*Figure 19.* **Memorization under various sampling conditions for `Gemma v1 (2B)`:** Barring a few conditions, the most prominent trend is that memorization reduces under multi-token prediction for various sampling conditions. Observe that the null and pause-conditioned models *do* produce some memorized output while their creativity was non-existent.

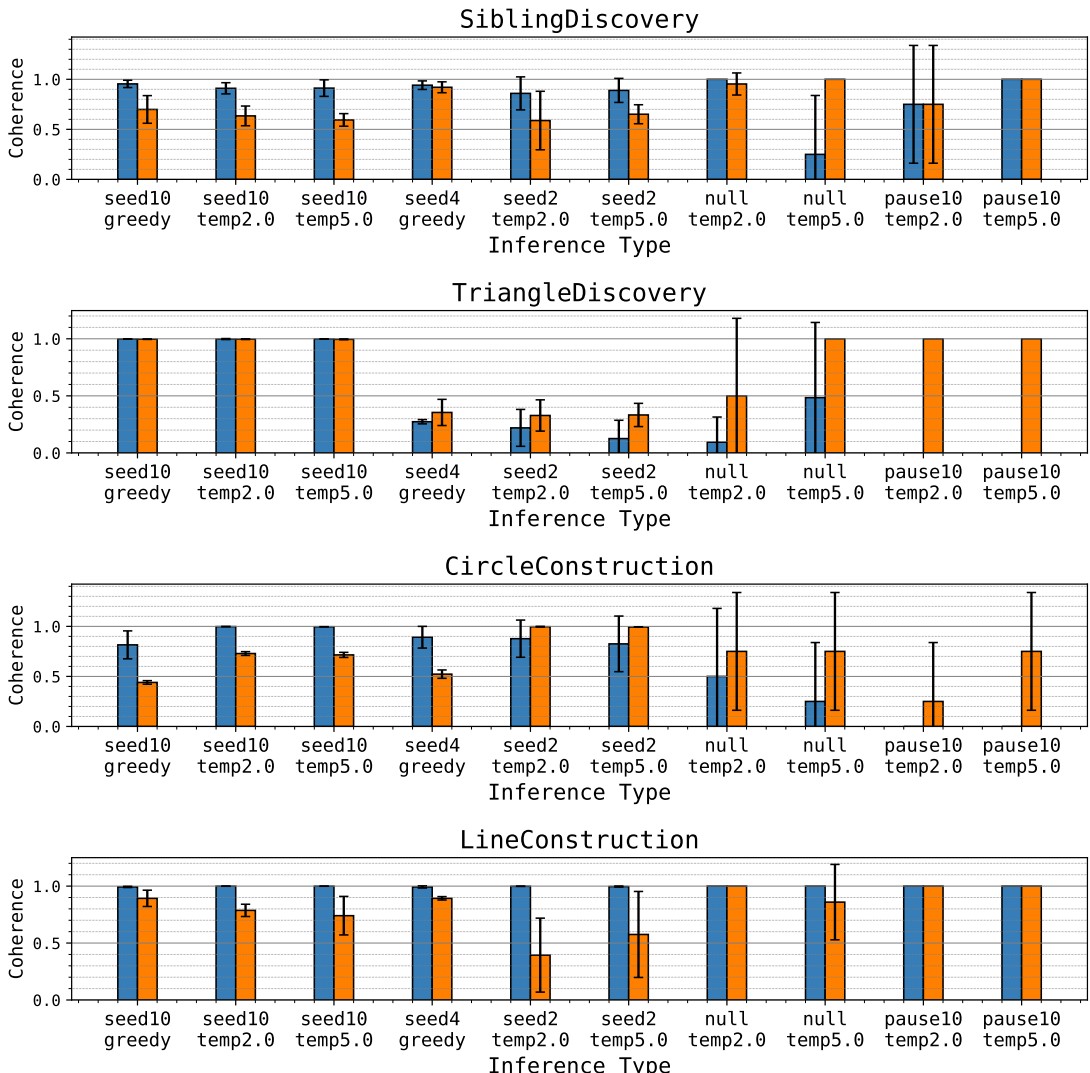

*Figure 20.* **Coherence under various sampling conditions for `Gemma v1 (2B)`:** Surprisingly, coherence of all models is high or at least noticeable, across various sampling conditions. This suggests that the low algorithmic creativity of the null-conditioned models in the previous plots arises from model collapsing to single original point.

# G. Additional experiments in `SEDD (90M)` vs. `GPT-2 (86M)`

## G.1. Ablation studies

In this section, we first provide additional ablation studies for `SEDD (90M)` vs `GPT-2 (86M)` with different training and dataset settings (Fig 21 and Fig 22).

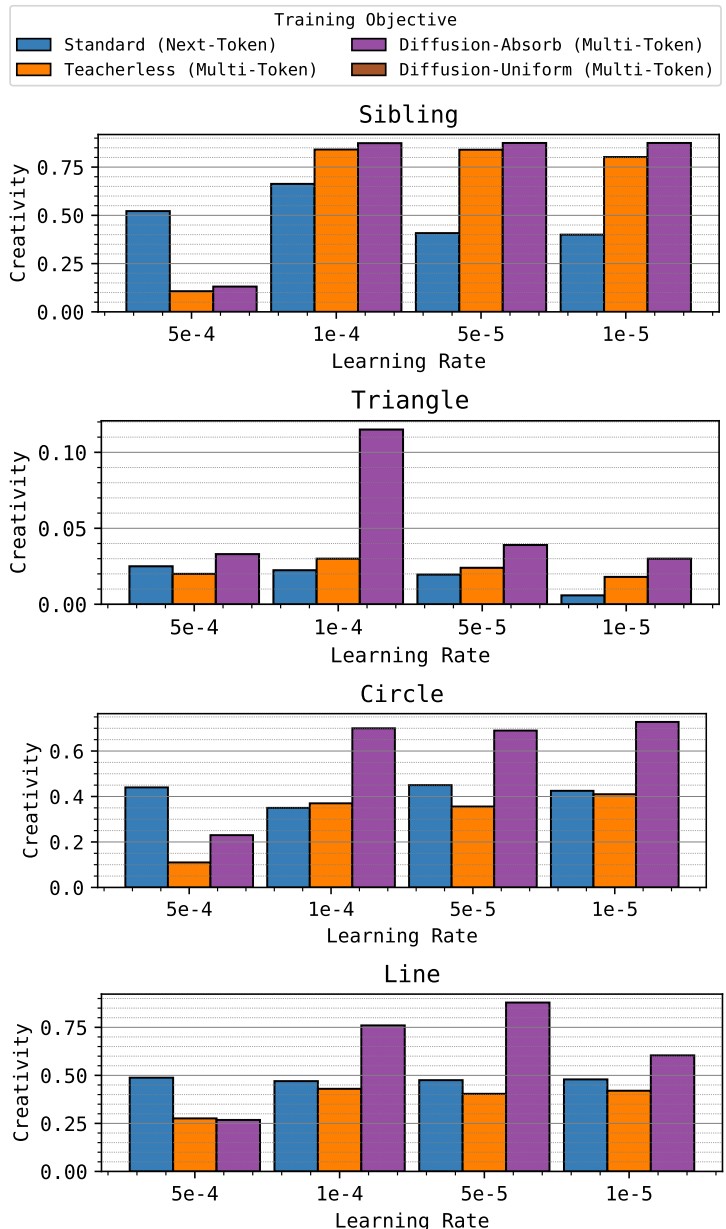

*Figure 21.* **Learning rates and algorithmic creativity for the `SEDD (90M)` model vs. `GPT-2 (86M)`:** MTP achieves higher algorithmic creativity than NTP when both are trained at their optimal learning rates. We use greedy decoding in this plot.

## G.2. Effect of seed length

We provide an ablation study on the seed length for NTP vs MTP on the `Sibling Discovery` task (Fig 23 (left)). We see that longer seeds lead to higher algorithmic creativity.

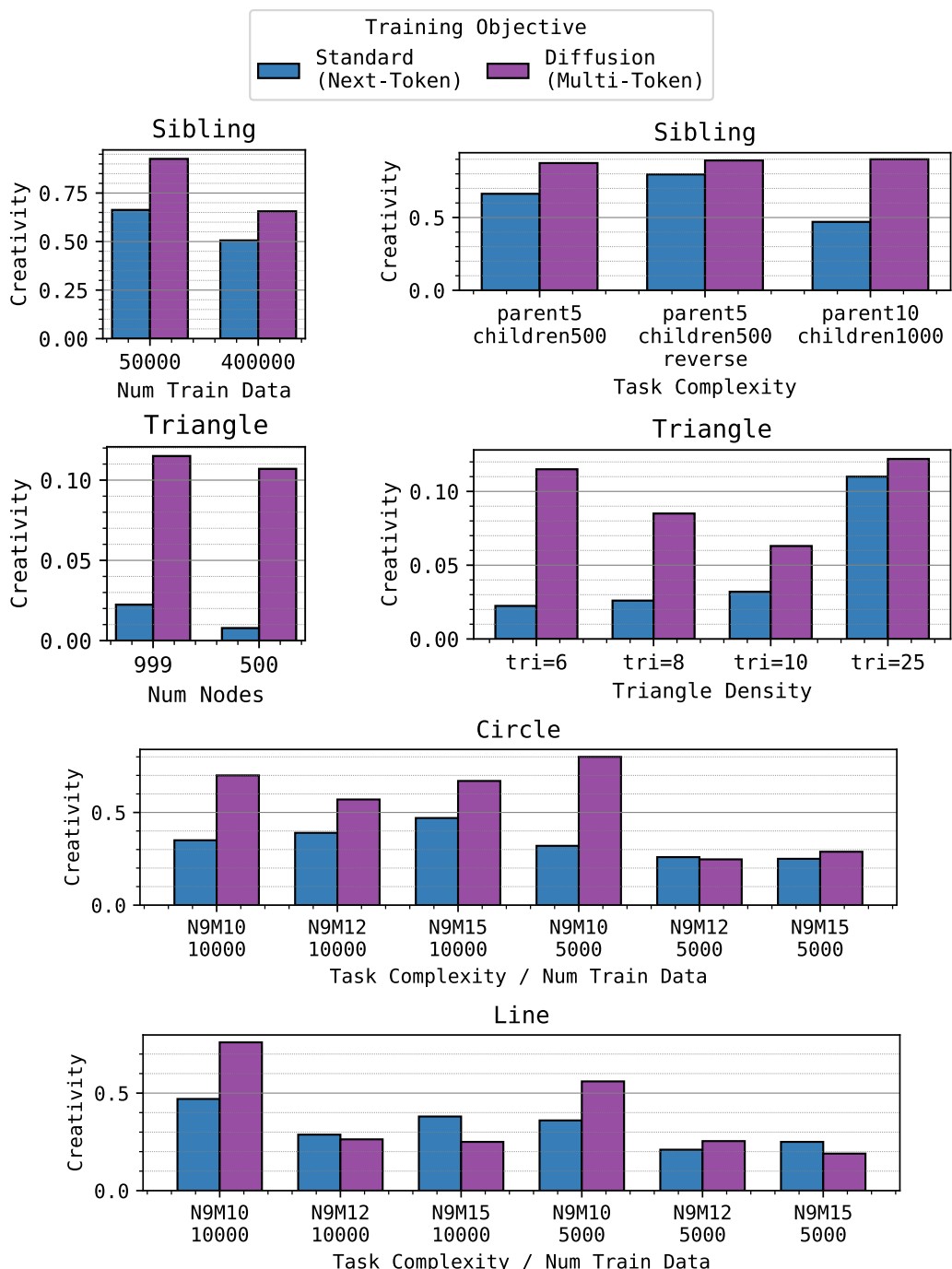

*Figure 22.* **Task complexity and algorithmic creativity of** `SEDD (90M)` **model vs.** `GPT-2 (86M):` MTP consistently outperforms NTP under varying task configurations, with some exceptions in the `Line Construction` and `Circle Construction` datasets. We use greedy decoding in this plot.

### G.3. Seed-conditioning for diffusion training

In Fig 23 (right), we apply seed-conditioning to the `SEDD (90M)` on the `Sibling Discovery` task. Unlike what we see on transformers, we do not observe improved creativity when applying seed-conditioning to diffusion training.

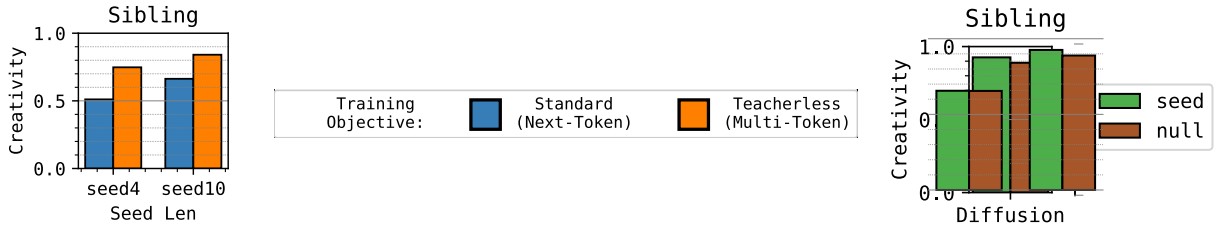

*Figure 23.* `GPT-2 (86M)` **Transformer achieves higher algorithmic creativity with longer seeds (top):** We report algorithmic creativity with seeds of length 4 and 10, with both `NTP` and teacherless `MTP` (with greedy decoding). **We do not see gains of seed-conditioning when it comes to diffusion training (right):** We report algorithmic creativity of the `SEDD (90M)` model with or without seed-conditioning.

### G.4. Effect of top-K sampling

Figure 24 reports the algorithmic creativity scores of NTP vs teacherless MTP training with seed-conditioning and top-$K$ sampling where K=50.

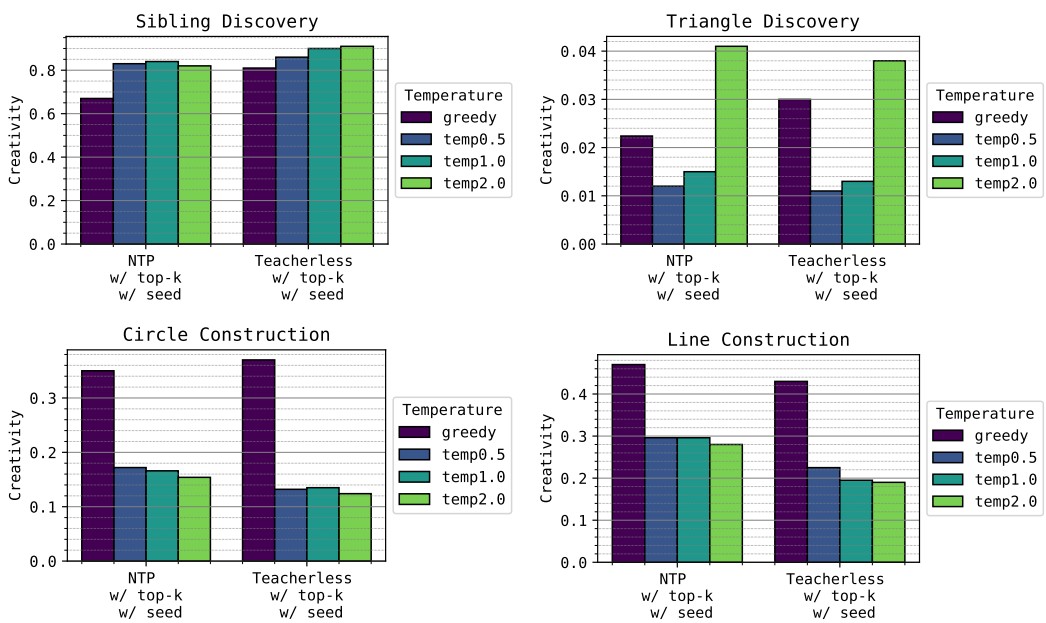

*Figure 24.* **Creativity scores for** `NTP` **vs teacherless** `MTP` **training with seed-conditioning + top-$K$ sampling where $K = 50$.**

### G.5. Format sensitivity for `Triangle Discovery`

Recall that our input format for `Triangle Discovery` follows the *edge list representation* of triangles (§D, Fig. 10). For instance, triangle ABC is represented as `AB, BC, CA`. This format explicitly lists the edges of the triangle, making it easier for the model to attend to edge-level patterns during learning.

We also experimented with an alternative *node-based representation*, where triangle ABC is represented more compactly as `ABC`, without making the edges explicit. We note in Fig 26 that the models are curiously sensitive to the way the triangles are formatted. Models trained on the node-based format perform equally badly with all training objectives, while the diffusion model outperforms `NTP` by a large margin with the edge list representation.

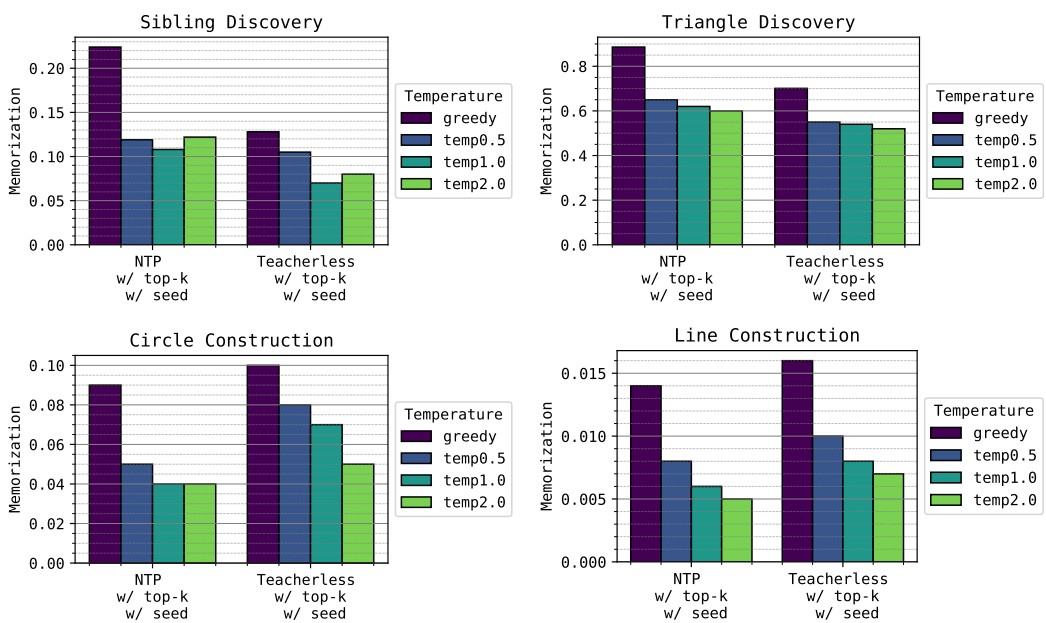

*Figure 25.* **Memorization scores for `NTP` vs teacherless `MTP` training with seed-conditioning + top-$K$ sampling where $K = 50$.**

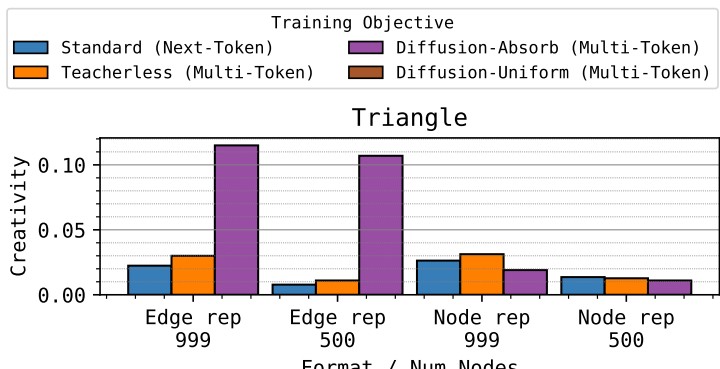

*Figure 26.* **Sensitivity to formatting of the sequence in `Triangle Discovery`:** We find that all our small models perform equally poorly with a node-wise representation of the input sequence, whereas there was a stark difference in performance with the edge-wise representation. We use greedy decoding in this plot.

### G.6. Additional ablations

Figure 28 reports the ablations on `GPT-2 (86M)` for (1) `NTP` vs `MTP`, (2) effect of seed conditioning, (3) effect of top-$K$, and (4) effect of temperature sampling.

## H. Additional experiments with medium-sized Transformer and SEDD

We replicate our `SEDD (90M)` and `GPT-2 (86M)` experiments on a larger model size ($\sim$400M parameters). In Fig 29, we see similar trends to the smaller model sizes (Fig 4).

## I. Decomposing creativity

Through following experiments on the `GPT-2 (86M)` model in the `Sibling Discovery` task, we try to understand the dynamics between two important quantities that affect algorithmic creativity: diversity/duplication and originality/memorization.

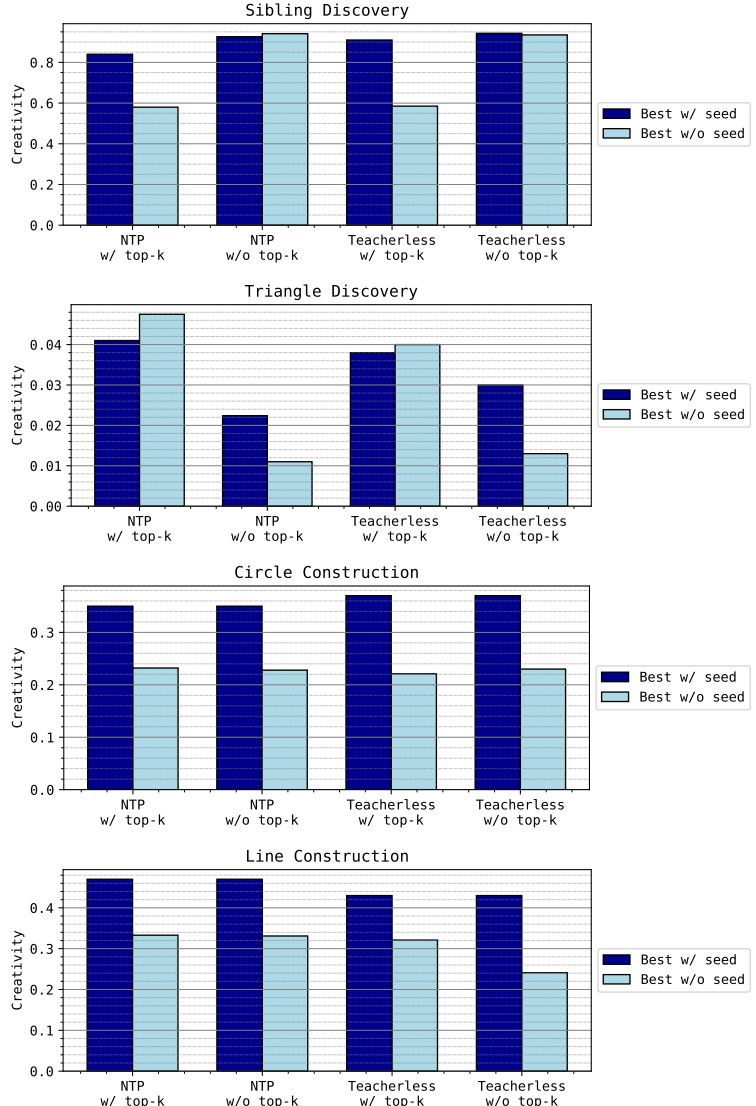

*Figure 27.* **Seed conditioning improves creativity.** "Best with seed" and "best without seed" stand for the best creativity score in each setting after we tune the sampling temperature from $\{0, 0.5, 1, 2\}$.

### I.1. Diversity score

Equation (1) defines our algorithmic creativity by rewarding samples that are both unique and novel. A higher score can be achieved either by enhancing diversity or by reducing memorization. In the following section, we examine this decomposition using the `Sibling Discovery` task. Formally, we define the diversity score as:

$$\hat{\mathrm{dv}}_N(T) = \frac{\mathtt{uniq}(\{\boldsymbol{s} \in T | \mathtt{coh}(\boldsymbol{s})\})}{|T|}. \tag{4}$$

We first demonstrate that creativity and diversity are not necessarily correlated, and next, that `MTP` particularly improves creativity (while achieving lower diversity than `NTP`). To show this, we report the algorithmic creativity and diversity scores along training in Fig 30. We see that for `NTP`, the diversity score keeps increasing and stays high, while algorithmic creativity increases in the first 10k steps and starts to decrease. For teacherless training, both scores increase throughout training. While the diversity of `MTP` is surprisingly lower than `NTP` throughout training, the creativity of `MTP` surpasses `NTP` at 20k steps.

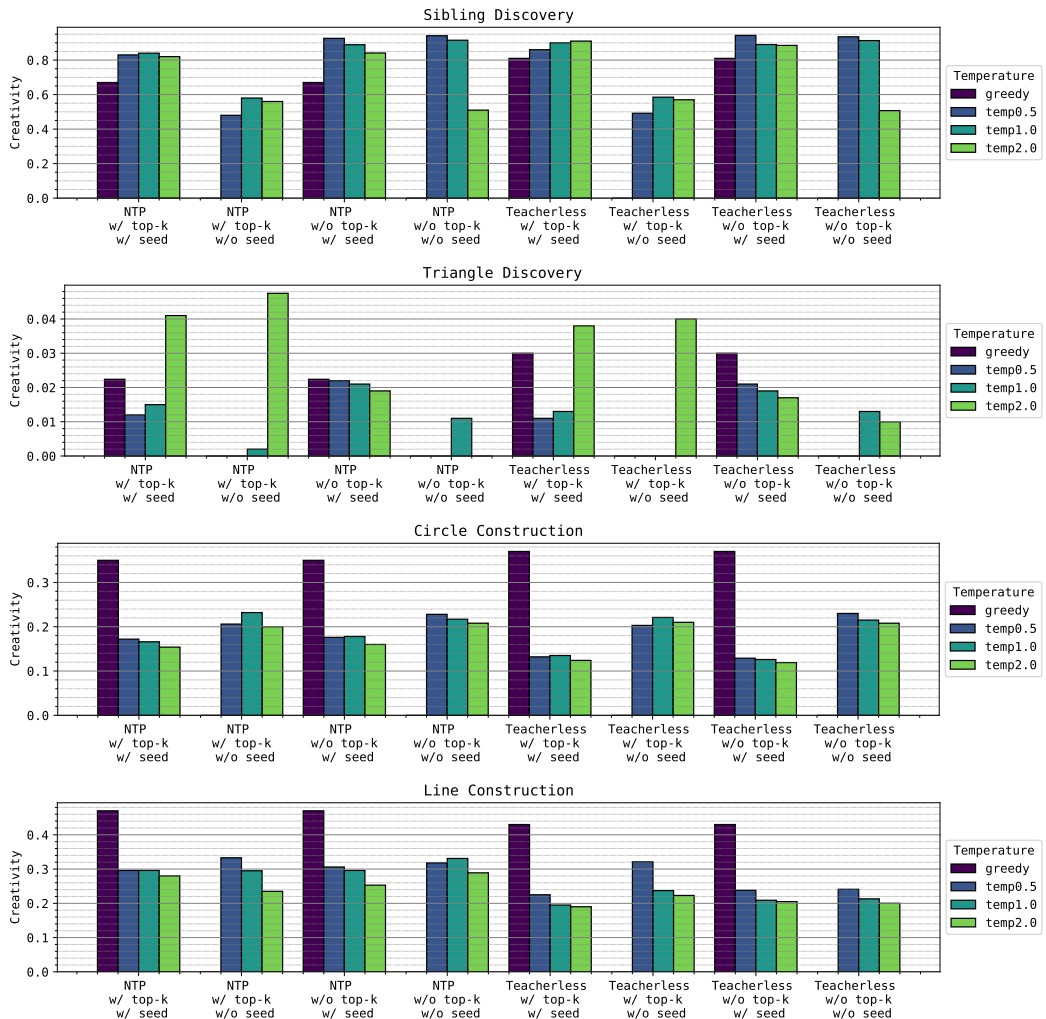

*Figure 28.* **Ablations on `GPT-2 (86M)` for (1) `NTP` vs teacherless `MTP` training, (2) effect of seed conditioning, (3) effect of top-K, and (4) effect of temperature sampling**

## I.2. Decomposing algorithmic creativity as diversity and memorization

Better creativity can be achieved either by enhancing diversity or by reducing memorization – we try to disentangle these factors in this section. In Fig 31, we plot the algorithmic creativity, diversity, and memorization scores at the checkpoint of best algorithmic creativity. We see that seed-conditioning contributes to higher diversity but does little to bring down memorization; however, teacherless `MTP` training contributes to higher diversity and also to reducing memorization. In Fig 30, we see that the best creativity and best diversity are not achieved at the same checkpoint.

## I.3. Data scaling for algorithmic creativity

How does algorithmic creativity change as we increase the amount of training data? Intuitively, more training data helps the model learn the true distribution, but also makes it harder to generate unseen samples (since the uncovered space becomes rarer). To understand this, we plot how models perform relative to a *theoretically expected* maximum algorithmic creativity. This is computed by assuming an oracle that samples a generated set $T$ (in Eq. (1)) uniformly with replacement from the true underlying distribution, and then computing algorithmic creativity Eq. (1. In Fig 32, we see that as we increase the training data (for a fixed underlying graph), the theoretically expected creativity decreases as expected, while the theoretically expected diversity stays the same (since this quantity does not care about being original with respect to the training set). Interestingly, as training data increases, `MTP` narrows the gap between `NTP` and the theoretically expected creativity and

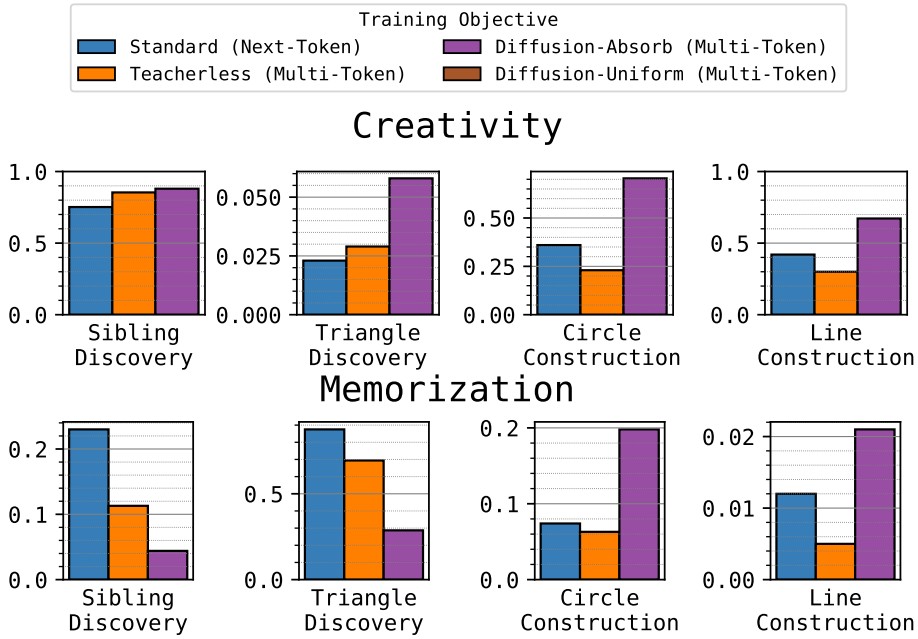

*Figure 29.* **On a medium-sized (∼400M) model, multi-token diffusion training improves algorithmic creativity from Eq 1 (top) on our four open-ended algorithmic tasks. We use greedy decoding in this plot.**

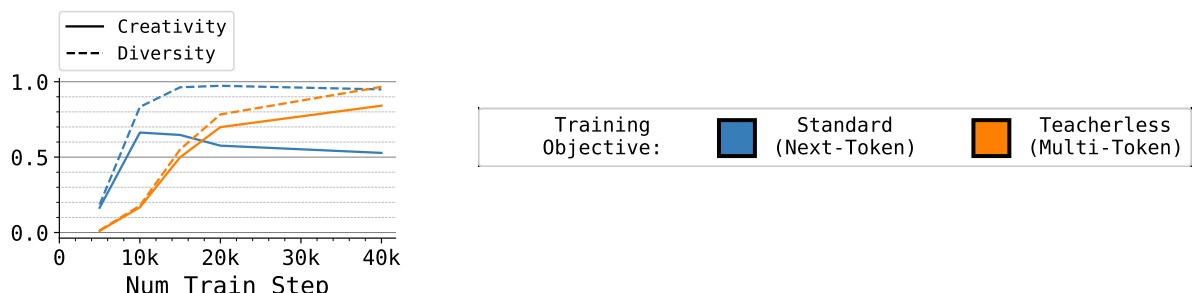

*Figure 30.* **Algorithmic creativity and diversity are not necessarily correlated, exhibiting distinct dynamics:** We find that `NTP` has a high diversity score through training, even higher than `MTP`. However, its algorithmic creativity reaches only a mediocre peak before descending, when `MTP` starts surpassing it. We use greedy decoding in this plot.

almost achieves the theoretically expected performance in the high data regime.

## J. Experiments on summarization

**Experimental Details.** In Table 3, we provide the hyperparameter details for the `GPT` models finetuned on both `XSUM` (Narayan et al., 2018) and `CNN/DailyMail` (Nallapati et al., 2016) for one epoch. We use a learning rate with linear warm up for $0.05$ of the total steps, followed by linear decay to $0$. To measure `Rouge` and `Self-Bleu`, we generate and average across 5 summarizations per document, on a test dataset $T$ of $250$ datapoints. We finetune our models with either the `NTP` objective (Eq 2) or the teacherless `MTP` objective (Eq 3), with equal weight to both.

To measure quality, we compute the average of `Rouge-1, Rouge-2, Rouge-L` as `Rouge`. For measuring diversity, we generate five different summaries per test example, and compute `Self-Bleu`. This computes average pairwise sentence `Bleu-2` scores with weights $(0.5, 0.5, 0, 0)$ on 1- and 2-tuples.

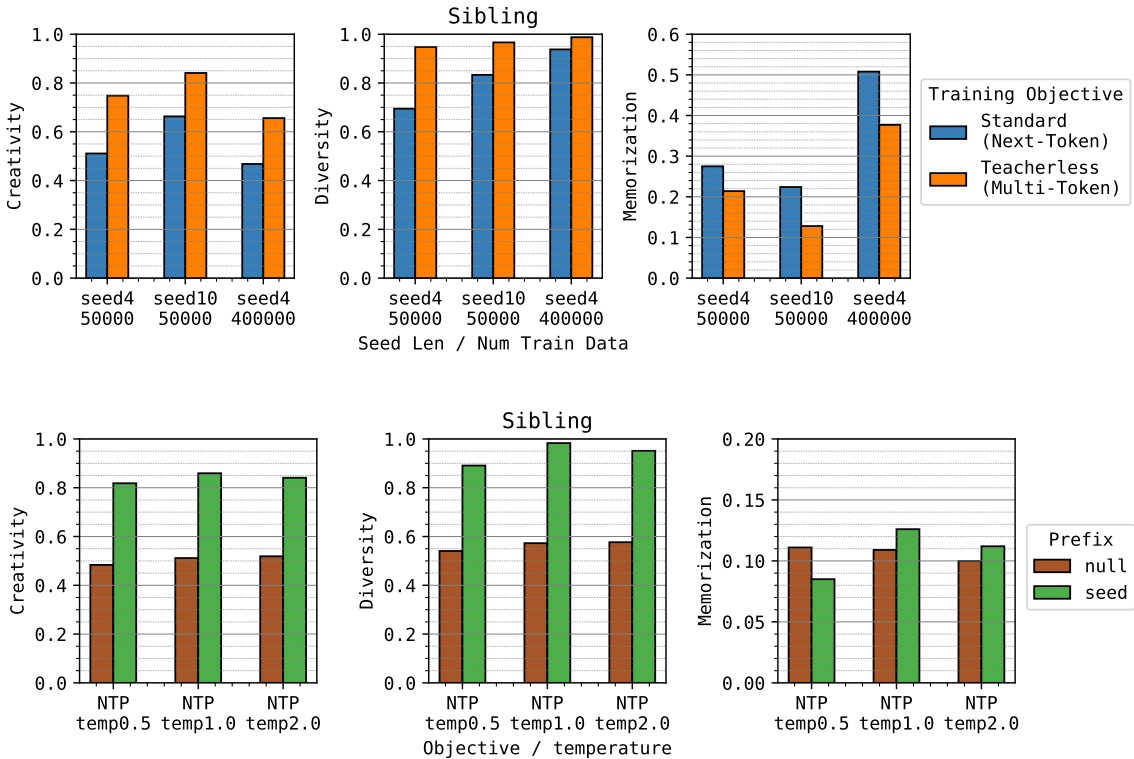

*Figure 31.* **Decomposition of algorithmic creativity for `GPT-2 (86M)` in `Sibling Discovery`:** We report algorithmic creativity, diversity and memorization at the checkpoint of best algorithmic creativity. We see that seed-conditioning contributes to higher diversity but does not help bring down memorization; teacherless training helps both diversity and in bringing down memorization.

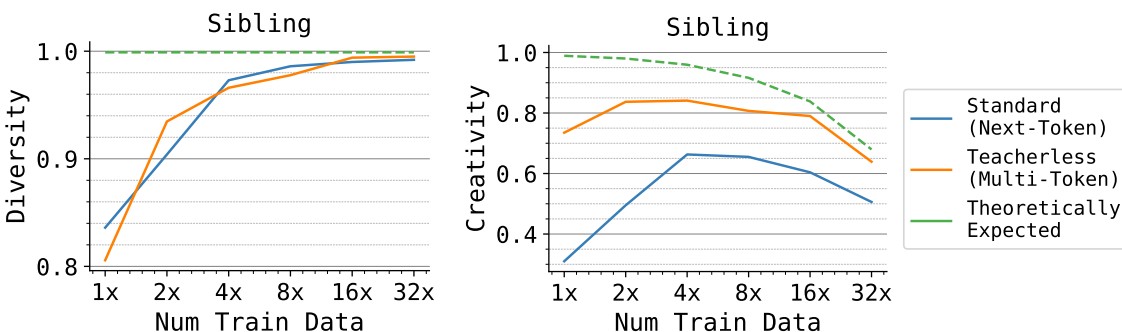

*Figure 32.* **Data scaling curve for algorithmic creativity and diversity:** As we increase the training data (for a fixed underlying graph), the theoretically expected maximum algorithmic creativity decreases as expected, while the theoretically expected maximum diversity stays the same. `NTP` tails to achieve the theoretically expected algorithmic creativity, while `MTP` almost achieves the theoretically expected performance at scale.

### J.1. Additional graphs for effect of multi-token training

Fig 33 shows the diversity and quality graphs on the smaller-sized `GPT-2` models on `XSUM`, and Fig 34 for `CNN/DailyMail`. While we consistently see improved quality from the multi-token model across the board, we don't see an increased diversity for fixed `Rouge` scores anymore.

*Table 3.* **Hyperparameter details for summarization experiments.**

| Hyperparameter | XSUM | CNN/DailyMail |
|---|---|---|
| Batch Size | 32 | 32 |
| Max. Learning Rate | $5 \times 10^{-5}$ | $3 \times 10^{-6}$ |
| Warmup Steps | 338 | 124 |
| Training Steps | 7778 | 2486 |
| Training Size | 248906 | 79552 |

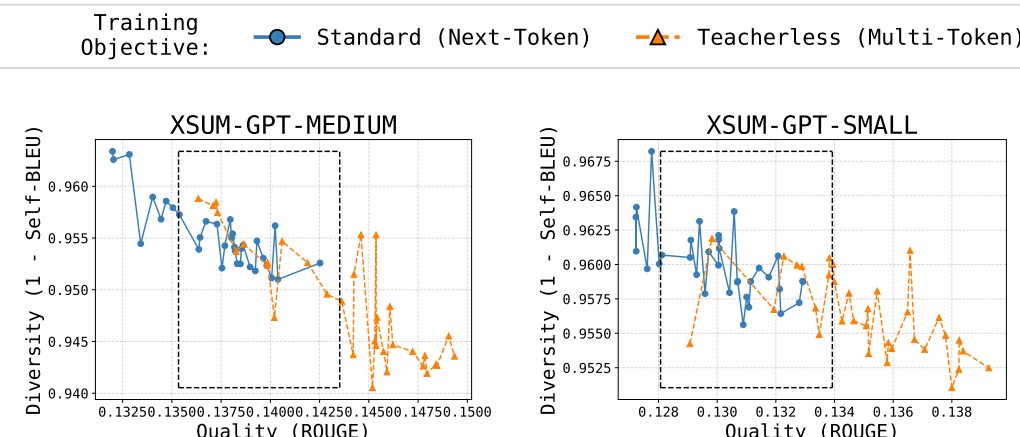

*Figure 33.* **Multi-Token Objective has no effect on diversity for smaller `GPT` models on `XSUM`.**

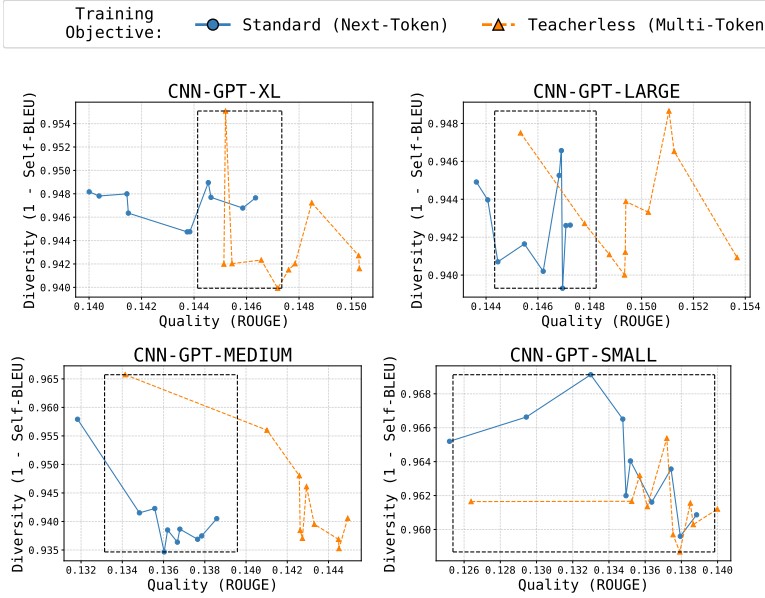

*Figure 34.* **Multi-Token Objective increases diversity for `GPT-L` and `GPT-M` but not for `GPT-XL` or `GPT-S` on `CNN/DailyMail`**

## J.2. Effect of seed-conditioning

We also conducted seed-conditioning experiments as described in §3.1. The seed strings we use are 10 randomly sampled uppercase characters from the English alphabet. We report the quality-diversity plots in Fig 35 (for next-token prediction on XSUM) and Fig 36 (for multi-token prediction on XSUM). As such, we do not find any changes in diversity, perhaps because this is not a sufficiently open-ended task.

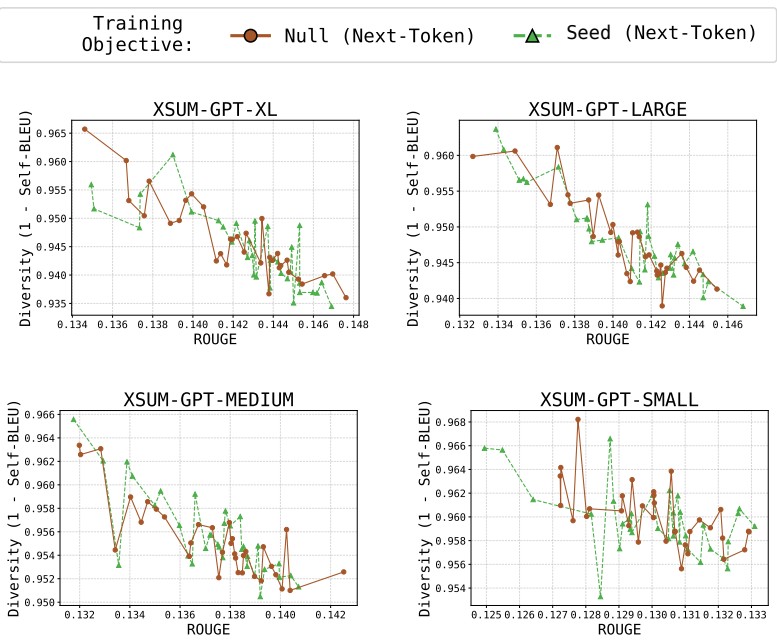

*Figure 35.* **Seed-conditioning has no effect on diversity for GPT models on XSUM summarization with next-token prediction.**

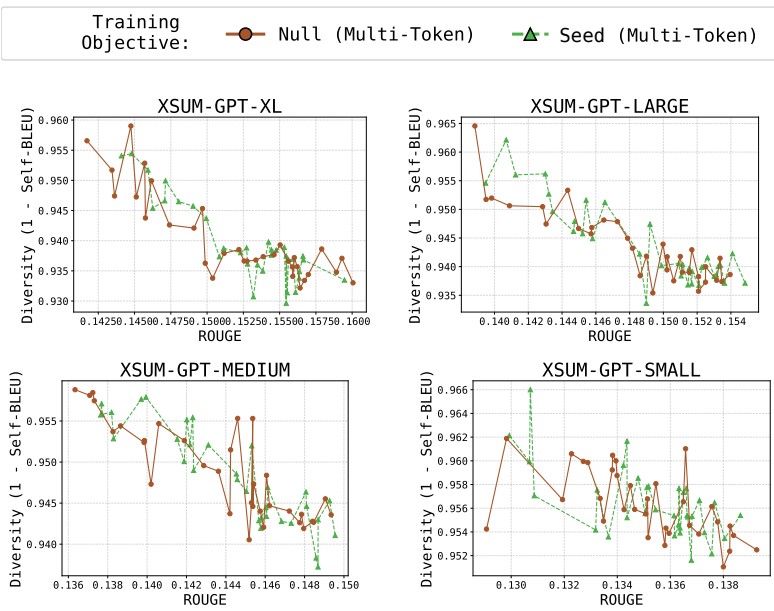

*Figure 36.* **Seed-conditioning has no effect on diversity for GPT models on XSUM summarization with multi-token prediction.**

## K. More related works

**Empirical studies of creativity in LLMs.** There is a long line of recent works that measure novelty and creativity of LLMs and LLM-assisted users. (Chakrabarty et al., 2024; Lu et al., 2024b) quantitatively evaluate and report that models vastly underperform under expert human evaluation against human writers. Zhang et al. (2024a) argue that finetuning methods such as RLHF and DPO, are limited when applied to creative humor-generation tasks. Likewise models like GPT4 and Claude currently underperform top human contestants in generating humorous captions. In poetry, Walsh et al. argue that there are certain characterstic styles that ChatGPT restricts itself to. Even assisted-writing can reduce diversity (Padmakumar & He, 2024) or produce bland writing (Mirowski et al., 2024). On the positive side, Si et al. (2024) report that LLMs surprisingly generate novel research ideas, although these are less feasible. Anderson et al. (2024) find that users tend to produce more divergent ideas when assisted by ChatGPT (although at a group level, ideas tend to homogenize). Another line of works (Wang et al., 2024a; Talmor et al., 2020; Zhong et al., 2024) has proposed algorithmic improvements involve creative leaps-of-thought for real-world tasks.

Other studies have proposed benchmarks for evaluating creativity. AidanBench (McLaughlin et al., 2024) and NoveltyBench (Zhang et al., 2025) evaluate LMs on their ability to produce diverse and coherent responses by penalizing repetition across generations. However, they do not measure originality relative to training data, leaving open whether outputs are genuinely novel or simply unseen paraphrases/recombinations. Zhao et al. (2024) evaluate LM creativity using the Torrance Tests of Creative Thinking, a standard in human psychometrics. Another line of work such as Alchemy (Wang et al., 2021), IVRE (Xu et al., 2022), and DiscoveryWorld (Jansen et al., 2024) present simulations with hidden facts and rules, requiring LMs to explore, hypothesize, and test through interaction. While these simulations focus on pretrained models rather than examining how training shapes creative capabilities, they serve as valuable and realistic benchmarks for assessing the role of creativity in scientific discovery.

Finally, we refer the reader to Franceschelli & Musolesi (2023) for a rigorous treatment of philosophical questions surrounding creativity in LLMs. We also a refer to Wang et al. (2024b) for a theoretical treatment of how to formalize subjectivity in creativity.

**The next-token prediction debate.** In support of next-token prediction, there are arguments (Shannon, 1948; 1951; Alabdulmohsin et al., 2024) that claim that language is captured by NTP with models even superceding humans (Shlegeris et al., 2022) at NTP. There are also theoretical results emphasizing the expressivity (Merrill & Sabharwal, 2024; Feng et al., 2023) and learnability (Malach, 2023; Wies et al., 2023) of autoregressive Transformers as long as there is a sufficiently long chain of thought.

**Multi-token training.** While multi-token methods employ diverse strategies, a common feature is their reliance on objectives that capture dependencies across entire sequences. Representative examples include teacherless training (Bachmann & Nagarajan, 2024; Monea et al., 2023; Tschannen et al., 2023) and independent output heads or modules (Gloeckle et al., 2024; DeepSeek-AI et al., 2024) or inserting a lookahead attention (Du et al., 2023). Another line of research is discrete diffusion models (Hoogeboom et al., 2021; Austin et al., 2021; Gong et al., 2023; Lou et al., 2023), which avoid strict left-to-right factorization by iteratively refining an entire sequence at multiple positions. There are other models as well, such as energy-based models (Dawid & LeCun, 2023) and non-autoregressive models or (Gu et al., 2018). Frydenlund (2024) report gains from non-autoregressive models on the path-star task (Bachmann & Nagarajan, 2024), while Hu et al. (2025) propose a novel method call the Belief State Transformer that shows gains on these tasks. While we only report the effect of teacherless training on our tasks due to its simplicity, we speculate that these methods would show similar (or better) results as teacherless training in our settings.

**Transformers and graph algorithmic tasks.** Graph tasks have been used to understand various limitations of Transformers in orthogonal settings. Bachmann & Nagarajan (2024); Saparov et al. (2024) report that Transformers are limited in terms of learning to search tasks on graphs, while Sanford et al. (2024) provide positive expressivity results for a range of algorithmic tasks that process an graph. These works differ from our study of combinational creativity since their graphs are provided in-context and the tasks have a unique answer. Other works (Schnitzler et al.; Yang et al., 2024a;b) study multi-hop question answering on a knowledge graph; however, this does not require planning.

**Diversity of generative models.** One line of work relevant to us in the history of generative models is RNN-based VAE for text data (Bowman et al., 2016). The motivation, like in our work, was to learn high-level semantic features rather than

next-token features with the hope of producing more novel sentences. However, this suffered from posterior collapse, where the model ignores the latent variable altogether inspiring various solutions (Yang et al., 2017; Goyal et al., 2017). Our results on seed-conditioning are also reminiscent of a line of work on exploration in reinforcement learning (RL), where it has been shown that adding noise to the policy model parameters enables more efficient exploration than directly adding noises to the output space (Plappert et al., 2017; Fortunato et al., 2017).

**Learning-theoretic studies of diversity in LLMs.** Various theoretical works provide rigorous arguments for how preventing hallucination and maximizing the model's coverage are at odds with each other in abstract settings (Kalai & Vempala, 2024; Kalavasis et al., 2024; Kleinberg & Mullainathan, 2024). We clarify that this tension does not apply in our concrete settings. In those abstract settings, the strings in the support can be arbitrary and adversarially chosen whereas, our strings are generated by a simple rule (which can be learned).

Another theoretical question underlying generative models is that the optimum of their objectives are attained at perfect memorization; yet they tend to produce novel examples e.g., this question has been posed for GANs in Nagarajan et al. (2018) and for diffusion in Nakkiran et al. (2024) (see "remarks on generalization") or Kamb & Ganguli (2024). Okawa et al. (2023); Kamb & Ganguli (2024) study the creativity ability of image diffusion models in combining various concepts to create an image. Unlike our setting, these do not require planning.

**Injecting input noise.** We note that the concept of injecting input noise into a Transformer has been explored before, but for other functions (such as quality, robustness (Hua et al., 2022; Jain et al., 2024) or efficiency (Wang et al., 2024c).), and in other forms (e.g., inducing Gaussian noise).

