# OpenReview forum: "Roll the dice & look before you leap: Going beyond the creative limits of next-token prediction"
_ICML.cc/2025/Conference — ICML 2025 oral_

### Official Review · Reviewer_16p5 · 2025-02-25

**Overall Recommendation:** 4

**Summary:**

The paper analyzes the effectiveness of multi-token prediction in open-ended algorithmic tasks that require combinatorial and exploratory creativity. It argues that transformers trained with next-token prediction struggle in these tasks, whereas transformers using multi-token prediction or diffusion models perform better.

**Claims And Evidence:**

The claim is well supported by experiments conducted across three types of architectures: the pre-trained Gemma, the SEDD diffusion model, and GPT-2, and four types of tasks.

**Essential References Not Discussed:**

N/A

**Experimental Designs Or Analyses:**

The proposed tasks are well designed, and drawing an analogy with the wordplay “What kind of shoes do spies wear? Sneakers.” helps me better understand their purpose. The authors effectively present the experimental results using appropriate plots and clear visualizations.

**Methods And Evaluation Criteria:**

While this paper does not introduce any new methods, I believe that designing new benchmark tasks is a valuable contribution. The authors also provide diversity and memorization scores, along with performance metrics, which I consider to be appropriate evaluation criteria. However, it would be better to define the memorization score in the main text, not in the caption in Figure 3.

**Other Comments Or Suggestions:**

N/A

**Other Strengths And Weaknesses:**

Overall, the paper is well written and well organized. I did not find any major drawbacks.

**Questions For Authors:**

I am very interested in the hash-conditioning experiment and have a few questions:
1. What exactly does a null prefix refer to? Does it mean that there is no additional tokens between the input and output?
2. From my understanding, the role of the random hash prefix is to retrieve a random sibling-parent triplet (or triangle, circle, etc.) from the input graph. Does this interpretation make sense?
3. If possible, could you run experiments where the transformer predicts the output while attending only to the hash prefix? This experiment would help clarify why the hash prefix enhances performance in transformer models while having minimal impact on diffusion models.

**Relation To Broader Scientific Literature:**

N/A

**Theoretical Claims:**

N/A

---

> ### Author Rebuttal · Authors · 2025-04-01
>
> Thank you for your time and effort in evaluating the paper! We are excited that you appreciate the analogy between wordplay and our tasks! We are also happy that you found (a) the tasks well-designed, (b) hash-conditioning interesting and (c) did not find any major drawbacks with the work!
>
> We respond to your questions below.
>
>
> > the role of the random hash prefix is to retrieve a random sibling-parent triplet (or triangle, circle, etc.) from the input graph. Does this interpretation make sense?
>
> Exactly! Intuitively, we view it as a random seed that guides the model in its random search over the graph to identify the siblings/triangles.
>
> For what it’s worth, more abstractly, we believe hash-conditioning may help via two mechanisms:
> - **More representation:** In softmax sampling, if the model has to maximize diversity, the model must maintain multiple diverse thoughts in its representations so that one of them can be sampled at output.  Hash-conditioning allows the model to fixate on just “one thought for a given hash”. This way the model can carefully represent one thought while being able to produce diverse thoughts for different hashes.
> - **Better planning:** Fixing the randomness ahead of time could allow the model to plan and co-ordinate multiple random decisions in tandem.
>
> > While this paper does not introduce any new methods,
>
>
> We wish to gently highlight that hash-conditioning is a novel method we put forth even if we only explore it on our algorithmic test bed. We believe this opens significantly new algorithmic possibilities for creativity in generative models e.g., could one pretrain language models with such hash strings (or by randomly inserting “hash tokens”) to encourage diversity?
>
> > What exactly does a null prefix refer to? … no additional tokens between the input and output?
>
> You’re right! A null prefix indicates there are no additional tokens between the input and output sequences. Sorry about the lack of clarity -- we will fix it!
>
> > If possible, could you run experiments where the transformer predicts the output while attending only to the hash prefix? This experiment would help clarify why the hash prefix enhances performance in transformer models while having minimal impact on diffusion models.
>
>
> If we understand correctly, we believe that one of our experimental setups already does this: specifically, teacherless training with hash-conditioning. Here, the model attends only to the hash prefix and some dummy tokens. We are curious if you have thoughts about the implications of this!
>
> > I also attempted to review Appendix D to find an example of the data, but it was not provided. It would be better to include an example for each task
>
> Thanks for the feedback! We will improve the clarity of the task data with more illustrated examples, but for now, we note that we provide one example for each task in Figure 1&2, and **the full description of dataset generation** in Appendix D.

---

> > ### Comment · Reviewer_16p5 · 2025-04-02
> >
> > Yes, your hashing-conditioning method is indeed novel. Regarding the additional experiment, I had assumed that even in teacherless training with hash-conditioning, the output tokens attend both the input graph and the hash prefix. I was just curious whether the transformer would still perform well if the output tokens attended only to the hash prefix. Please let me know if I am wrong. Regarding the last question, I was simply interested in seeing how the actual input and output are represented in text for each task.
> >
> > Overall, the paper is well-organized, and the experiments are conducted appropriately. Since my questions are well addressed, I increase my score from 3 to 4.
> >
> > Update: Thank you for the illustration!

---

> > > ### Author Response · Authors · 2025-04-08
> > >
> > > We appreciate your updated assessment of our paper!
> > > Regarding the dataset examples: we provided a graphical illustration of our datasets at this anonymous link: https://gist.github.com/lm-creativity-25/997631eae2237c09cbf9dfe3d41e9e00 The exact details (e.g., tokenization) are provided in the appendix. Also, for Sibling and Triangle, the graph is in-weights rather than in-context – mirroring how pretraining data is derived from the underlying real-world knowledge.

---

### Official Review · Reviewer_gG8C · 2025-03-14

**Overall Recommendation:** 4

**Summary:**

This paper focuses on addressing the issue of insufficient creativity in traditional next-token prediction (NTP) when applied to open-ended algorithmic tasks. The authors have designed a series of minimalistic algorithmic tasks (such as Sibling Discovery, Triangle Discovery, Circle Construction, and Line Construction) to simulate real-world scenarios that require creative thinking. To this end, the paper proposes a multi-token prediction approach, primarily realized through Teacherless Training and Discrete Diffusion Models, and introduces a Hash-Conditioning strategy to enhance the diversity of generation. Experimental results demonstrate that, compared to traditional NTP methods, the multi-token approach exhibits significant advantages in boosting creativity and reducing the model's tendency to memorize and reiterate.

**Claims And Evidence:**

Yes

**Essential References Not Discussed:**

N/A

**Experimental Designs Or Analyses:**

Yes

**Methods And Evaluation Criteria:**

make sense

**Other Comments Or Suggestions:**

Suggestions:

(1) Expand the scope of experiments to other practical tasks to assess the generalizability of the proposed method. (2) Incorporate additional evaluation metrics to measure creativity, and conduct human studies to evaluate the effectiveness. (3) Provide a more detailed demonstration of how the global perception capability of MTP (Multi-Token Prediction) contributes to the task at hand, supported by use case studies.

**Other Strengths And Weaknesses:**

Strength:

(1) The investigation into the issues surrounding NTP holds substantial theoretical significance and practical application prospects. (2) Controlled experimental design: By devising simplified algorithmic tasks, the study has clearly quantified two metrics—creativity and memorized repetition—thereby facilitating the demonstration of the method's advantages.

Weakness:

(1) The mechanism underlying the role of hash-conditioning has not been clearly elucidated. (2) Although the minimalistic tasks in the paper facilitate quantification and analysis, there exists a gap between these tasks and real-world applications, casting doubt on their generalizability. (3) The performance of large-scale NTP models (only Gemma-2b and GPT2-86M were used in the study) in this context, as well as the effectiveness of reasoning enhancement methods on these tasks, has not been evaluated.

**Questions For Authors:**

See **Suggestions**

**Relation To Broader Scientific Literature:**

Key contributions of the paper related to the topic of Multi-token prediction of LLM and Next-token prediction.

**Theoretical Claims:**

Yes

---

> ### Author Rebuttal · Authors · 2025-04-01
>
> Thank you for your time & insightful feedback! We address your points below:
>
> ### Major points
>
> > The mechanism underlying the role of hash-conditioning has not been clearly elucidated
>
> Here's our current intuition which we'll add to the paper.
>
> Recall that hash-conditioning explicitly injects randomness in the input; in softmax sampling, randomness is elicited from model output.  **Hash-conditioning may then help via two mechanisms:**
>
> - **More representational power:** In softmax sampling, if the model has to maximize diversity, the model must maintain multiple thoughts in its representations so that one of them can be sampled at output.  Hash-conditioning allows the model to fixate on just “one thought for a given hash”. This way the model can carefully represent one thought while being able to produce diverse thoughts for different hashes.
> - **Better planning:** Fixing randomness ahead of time allows the model to plan and co-ordinate multiple random decisions in tandem.
>
> We emphasize that hash-conditioning is a novel approach proposed as part of our broader exploration of creativity (as Reviewer `Lrgc` and `16p5` remark as very interesting). **Thus,
> we invite you to view hash-conditioning as a novel algorithmic contribution (in a larger paper) with reasonable intuition behind it,** opening up interesting future directions (such as verifying the above mechanisms & generalizing it).
>
> > Although the minimalistic tasks in the paper facilitate quantification and analysis, there exists a gap between these tasks and real-world applications, casting doubt on their generalizability
>
> It is absolutely right to acknowledge the gap between our very simple tasks and ambitious real-world tasks.
>
> As the first step, we provided summarization as a practical example demonstrating real-world applicability.
> However, the challenge with generalizing real-world tasks is that there is no pre-existing benchmark where creativity, originality and diversity are even quantifiable e.g., it’s **impossible** to judge originality when the dataset is the whole of the internet. This would require a non-trivial effort that is worth a few other papers’ work!
>
> Regardless, we’re happy that you (and also the other reviewers) acknowledge the value in studying our algorithmic tasks: they help us cleanly measure creativity/diversity/originality and analyze how different training methods can affect creativity.
>
> > The performance of large-scale NTP models (only Gemma-2b and GPT2-86M were used in the study) in this context,..., has not been evaluated.
>
> We agree that an ideal study would study a range of larger models.
> - But it’s worth noting that, **for our very minimal dataset sizes/complexities, the 2B scale is very, very large!** (Indeed, Reviewer Lrgc notes that these models are “reasonable”).
> - Additionally, we have observed limited absolute performance gain with increasing model size.
> - **We have new experimental results that reproduce the benefit of diffusion models over NTP at the 400M parameter scale (the largest open-sourced one we can find)**. We will add detailed scaling plots for each task in the final version of the paper.
>
> > as well as the effectiveness of reasoning enhancement methods on these tasks
>
> This is a completely valid question that we are curious about too. However, there is doubt as to whether “reasoning enhancement” methods would translate to “creativity enhancement". This is because we not only care about “correctness/coherence of planning/reasoning”, but the “coherence **+ originality + diversity** of a plan”. We kindly refer you to **our response to Reviewer Lrgc (first quoted question)** where we provide three arguments substantiating our point. These are profound questions that require multiple papers of future research.
>
> ---
> ### Other points
> > Incorporate additional evaluation metrics... human studies...; demonstration of how the global perception capability of MTP (Multi-Token Prediction) contributes to the task... use case studies.
>
> Thank you for these wonderful suggestions!
> Evaluation of creativity in the wild is hard and requires careful consideration worth the effort of a whole new paper.
> The current creativity metric we chose, Self-BLEU, is widely used as a diversity metric in language generation benchmarks [1]. It quantifies diversity by measuring BLEU scores across outputs generated by the same model, thus showing the distinctiveness among outputs. We are also exploring other diversity metrics (such as distinct n-gram).
> Human evaluations are beyond our current scope & expertise but remain a key future direction for the community.
>
> We will certainly keep these in mind for future follow-ups!
>
> [1] Texygen: A Benchmarking Platform for Text Generation Models
>
> ---
>
> **We sincerely hope we've addressed your key concerns (model size & other reasoning methods) in a way that allows you to re-evaluate the paper in more positive light, thank you!**

---

### Official Review · Reviewer_Lrgc · 2025-03-14

**Overall Recommendation:** 5

**Summary:**

In this work, the authors aim to study the failure of the next-token prediction (NTP) objective at open-ended creative tasks, where the goal is to generate a diversity of outputs satisfying some constraint. Motivated by the fact that many such tasks require learning to form a latent plan that is not captured from learning the distribution one token at a time, they hypothesize that multi-token prediction (MTP) objectives would be more suited to open-ended learning.

Motivated by work in the creativity cognitive science literature, the authors design a set of 4 algorithmic tasks whose distribution they aim to model. They define the creativity score as the number of unique non-memorized outputs that satisfy the task-dependent constraint.

The authors consider 2 MTP objectives from the literature for evaluation against NTP on these tasks: teacherless training and discrete diffusion. They consider 2 sets of models: Gemma 2b v1 models for NTP vs teacherless training, and an additional ~90m parameter setup for NTP vs Diffusion (and teacherless).

The authors find that across tasks and models, NTP leads to low creativity and high memorization, whereas MTP objectives consistently exhibit higher creativity score. The authors additionally find that a novel training method (hash-conditioning), wherein a model is trained with the hash of the sequence it is supposed to model as a prefix and at inference conditioned on a novel hash, consistently leads to higher creativity, even in the NTP setting.

**Claims And Evidence:**

All claims in the paper are supported by adequate evidence.

**Essential References Not Discussed:**

None found.

**Experimental Designs Or Analyses:**

The models used were reasonable, and all experiments with the 2b model were performed 4 times with variation plotted in the charts. The appendix contains detailed hyperparameter studies.

**Methods And Evaluation Criteria:**

The paper uses a set of simple algorithmic tasks to test for the failure cases of the NTP objective. While I am usually quite wary of toy tasks, here its actually quite useful to have a simplified setting. This allows the authors to measure creativity where, in natural language scenarios, this would be challenging (exact uniqueness would not work because of superficial difference, string distance would not capture conceptual differences, and metrics based on embedding distances are less interpretable and sensitive to choice of representation).

I mostly agree with the authors that this task is a good failure test of NTP prediction: models failing on these tasks at a small scale can reasonably be expected to fail at larger scale. A counterargument might be that large-scale models exhibit more capabilities than small-scale ones, but the experiments at 2b scale with pretrained models somewhat mitigate this concern. A second counterargument might be the existence of chain-of-thought capabilities in large models: using such chains of thought the model might be able to capture the latent plan needed to produce creative outputs. This is shortly discussed in the appendix of the paper and warrants its own followup investigation, nevertheless I think it is outside the scope of the current paper.

**Other Comments Or Suggestions:**

* Sec 3.1: typo “from an prompt-free autoregressive” -> “ from a prompt-free autoregressive”
* Sec 5: typo  “We defer discussion of theoretical studis of diversity”

**Other Strengths And Weaknesses:**

* This paper is exceptionally well-written, and the text addresses most questions that come up while reading;
* This paper is very timely, given the growth in interest in MTP methods recently;
* The tasks are quite elegant models of creative tasks and the relationship to combinatorial and exploratory creativity is well-drawn;
* The larger-scale tasks might require more explanation and exposition (why Self-Bleu as an evaluation metric?) but unfortunately space is lacking. This could make for a good follow-up study.
* The result on hash-conditioning is very interesting and should be validated with larger-scale empirical work; how usable is it in practice?
* Related work is extensive and presents an overview of the creativity literature, the MTP literature, the literature on strengths and weaknesses of NTP, and the theoretical literature on creativity in models.

**Questions For Authors:**

Question 1: Fig 5 Temperature 2.0 might be too much for Gemma, did you try lower temperatures (1.0)?

**Relation To Broader Scientific Literature:**

The relation to the broader literature is excellent, with the main paper detailing the closest works with algorithmic tasks, and the supplementary related work section delving in studies of LLM creativity in more realistic settings (notably creative writing) as well as multi-token objective in contemporary large-scale training (such as the work of Gloeckle et al 24 or the training of DeepSeek V3). The relationship between the teacherless objective and these large-scale efforts might be worth discussing in the main text.

The reference of the classical work of Boden as an inspiration for the tasks is most welcome.

**Theoretical Claims:**

No theoretical proofs in this paper.

---

> ### Author Rebuttal · Authors · 2025-04-01
>
> Thank you for your detailed and encouraging feedback. We are happy that you find our approach elegant and timely given how it is near-impossible to rigorously measure creativity in the real-world. We are also pleased that you find (a) our proposed hash-conditioning approach interesting (b) our claims well-supported, the model sizes reasonable and (c) our related work discussion extensive.
>
> ### Reg. CoT, RL etc.,
>
> > A second counterargument: using chains of thought… to capture the latent plan needed to produce creative outputs
>
> We _completely_ agree with your counterargument (and glad you acknowledge this to be out of scope of this paper!).
> Nevertheless, there are some profound doubts that arise when considering CoT in creative planning tasks which differ from standard reasoning tasks:
>
> 1. **Current paradigms like CoT, prompting, RL, instruct-tuning -- which are evaluated on math/coding tasks -- are purely designed for one goal: *coherence/correctness of a single, generated plan.*** None are explicitly designed for (i) diversity across various generations and (ii) originality of a generation compared against the training set. Importantly, it is highly unclear how to redesign these paradigms (the rewards, the prompts etc.,) to optimize originality or diversity!
>    - Arguably, the “SFT on iid samples from a distribution” setup — despite being less fancy — seems to be the most natural way to teach a model to produce diverse & original outputs.
> 2. **Human-generated data rarely contains the explicit, step-by-step trace behind creative thought.** For example, authors rarely document their internal thought processes while conceptualizing creative works like a research paper or a clever olympiad problem. **Given that the creative process is highly complex compared to standard CoT use-cases,** (see below!) to what extent can a model produce such traces with originality & diversity?
> 3. Even if one may hope such CoT emerges, **fundamentally, it is unclear if CoT traces even exist for many creative tasks.**
>    - For instance, in the triangle discovery task, is there a CoT-style approach for maximizing creativity? Would that involve laboriously enumerating all triplets in the graph as CoT and filtering them out? Is such an approach even scalable in real-world tasks? What would the CoT trace for generating all possible original, and diverse completions for “a horse walked into the bar” look like?
>    - In contrast, we speculate that in many real-world tasks, creative _leaps_-of-thought come from "quick, internal, latent heuristic leaps" that do _not_ occur in token space. (Our non-CoT setup seems better suited for modeling this.)
>
> Evidently, these are profound questions that future research must tackle.
>
> ---
>
> > The larger-scale tasks might require more explanation and exposition (why Self-Bleu as an evaluation metric?) but unfortunately space is lacking.
>
> We will improve this discussion!  We choose Self-BLEU as it’s widely used as a diversity metric in language generation [1]. It measures BLEU scores across outputs generated by the same model to measure distinctiveness among outputs.
>
> A full fledged future investigation would certainly strengthen our findings.
>
> [1] Texygen: A Benchmarking Platform for Text Generation Models
>
> > The result on hash-conditioning is very interesting and should be validated with larger-scale empirical work; how usable is it in practice?
>
> Absolutely — validating hash-conditioning at scale would have  great impact! **Currently, the roadblock is large-scale evaluation: we need _open-ended_ benchmarks where diversity & originality can be precisely measured alongside correctness/coherence.** While there are evaluations of creativity like in [2] [3] [4] [5], unfortunately, these either use debatable proxies (e.g., n-grams, sentence embeddings) or only measure the quality of a standalone generation without grounding to the training data - none of them are accurately evaluating originality and creativity.
>
> As for the algorithm itself, there are obvious ways to extend hash-conditioning to practice e.g., allocate a reserve of “hash-tokens”, and pretrain/finetune with such hash-tokens inserted in the model. We hope that our paper can inspire an active exploration of such ideas & also settle the benchmark challenge above.
>
> [2] Hu et al., The Belief State Transformer
>
> [3] Lu et al., Quantifying linguistic creativity of language models via systematic attri-bution of machine text against web text
>
> [4] Peeperkorn et al., Is temperature the creativity parameter of large language models?
>
> [5] Si et al., Can LLMs generate novel research ideas?  A large-scale human study with 100+NLP researchers
>
> > Fig 5 Temperature 2.0 might be too much for Gemma, did you try lower temperatures (1.0)?
>
> Good point; we in fact started with lower temperatures which only performs equivalently if not worse in terms of creativity, which is why we moved to exploring larger temperatures. We will make sure to add.

---

### Decision · Program_Chairs · 2025-05-01

**Decision:**

Accept (oral)

**Comment:**

The paper studies multi-token prediction for creativity in open-ended algorithmic tasks, introduces a novel testbed, and proposes hash-conditioning. Reviewers appreciate the novel and well-designed testbed, the rigorous quantification of creativity, strong empirical results showing MTP benefits, and the timely topic. However, initial questions were raised about clarifying hash-conditioning, the generalizability of tasks to real-world, and evaluating larger NTP models/reasoning methods. Overall, the paper is recommended for acceptance, as the authors' comprehensive rebuttal addressed concerns, confirming the work's significant contribution.